# Impact of the Ozone Monitoring Instrument Row Anomaly on the Long-term Record of Aerosol Products

Omar Torres[1], Pawan K. Bhartia[1], Hiren Jethva[2,1], Changwoo Ahn[3,1]

[1]Atmospheric Chemistry and Dynamics Laboratory, NASA Goddard Space Flight Center, Greenbelt, MD, 20770, USA
[2]GESTAR/Universities Space Research Association, Columbia, MD, 21046, USA
[3]Science Systems and Applications, Inc., 10210 Greenbelt Road, Lanham, Maryland 20706

*Correspondence to*: Omar Torres (Omar.O.Torres@nasa.gov)

## Abstract

Since about three years after the launch the Ozone Monitoring Instrument (OMI) on the EOS-Aura satellite, the sensor's viewing capability has been affected by what is believed to be an internal obstruction that has reduced OMI's spatial coverage. It currently affects about half of the instrument's sixty viewing positions. In this work we carry out an analysis to assess the effect of the reduced spatial coverage on the monthly average values of retrieved aerosol optical depth (AOD), single scattering albedo (SSA), and the UV Aerosol Index (UVAI) using the 2005-2007 three-year period  prior to the onset of the row anomaly.

Regional monthly average values calculated using viewing positions 1 through 30 were compared to similarly obtained values using positions 31 thru 60 with the expectation of finding close agreement between the two calculations. As expected, mean monthly values of AOD and SSA obtained with these two scattering-angle-dependent sub-sets of OMI observations agreed over regions where carbonaceous or sulphate aerosol particles are the predominant aerosol type. Over arid regions, however, where desert dust is the main aerosol type, significant differences between the two sets of calculated regional mean values of

AOD were observed. As it turned out, the difference in retrieved desert dust AOD between the scattering-angle dependent observation subsets was due to the incorrect representation of desert dust scattering phase function. A sensitivity analysis using radiative transfer calculations demonstrated that the source of the observed AOD bias was the spherical shape assumption of desert dust particles. A similar analysis in terms of UVAI yielded large differences in the monthly mean values for the two sets of calculations over cloudy regions. On the contrary, in arid regions with minimum cloud presence, the resulting UVAI

monthly average values for the two sets of observations were in very close agreement.  The discrepancy under cloudy conditions was found to be caused by the parameterization of clouds as opaque Lambertian reflectors. When properly accounting for cloud scattering effects using Mie Theory, the observed UVAI angular bias was significantly reduced. The analysis discussed here has uncovered important algorithmic deficiencies associated with the model representation of the angular dependence of scattering effects of desert dust aerosols and cloud droplets. The resulting improvements in the handling

of desert dust and cloud scattering have been incorporated in an improved version of the OMAERUV algorithm.

## 1.0 Introduction

The Ozone Monitoring Instrument (OMI) on the EOS-Aura satellite has been circling the Earth on an ascending orbit for over a decade since its launch in July 2004 [*Levelt et al.,* 2006]. The EOS-Aura spacecraft flies in formation with the Aqua, CALIPSO, CloudSAT and, until recently, PARASOL platforms in the A-train satellite constellation. OMI's hyper-spectral radiance measurements (270-500 nm) cover a 2400 km west-to-east across-track swath. Observations at sixty viewing positions or rows provide daily global coverage. These observations are used in retrieval algorithms to derive concentration of total ozone [*McPeters et al.,* 2015]; $NO_2$ and $SO_2$ [*Li et al.,* 2013; *Boersma et al.,* 2011; *Krotkov et al.,* 2016]; $CH_2O$ [*González et al.,* 2015] as well as aerosol properties [*Torres et al.,* 2007; 2013] and clouds [*Acarreta et al.,* 2004; *Joiner and Vasilkov,* 2006].

The OMI near UV algorithm (OMAERUV) uses observations at 354 and 388 nm to derive aerosol optical depth (AOD) and single scattering albedo (SSA) in addition to the qualitative UV Aerosol Index (UVAI) [*Torres et al.,* 2007]. The algorithm, based on TOMS (Total Ozone Mapping Spectrometer) heritage, takes advantage of the interaction of molecular scattering and particle absorption in the UV to detect and quantify absorption properties of UV-absorbing particulate such as carbonaceous, desert dust and volcanic ash aerosols [*Torres et al.,* 1998]. The retrieval algorithm relies on forward calculations of angle-dependent upwelling near UV radiances whose accuracy depend on the correct characterization of both molecular and particle scattering. Aerosol scattering depends on particle size, shape and composition. In the OMAERUV algorithm, the aerosol scattering phase function is calculated using Mie Theory, which applies only to spherical particles. Erroneous scattering phase function characterization may produce large errors in retrieved AOD and SSA values [*Gassó and Torres, 2016*]. OMAERUV uses external ancillary data from other A-train sensors for characterization of aerosol type and information on aerosol layer height [*Torres et al.,* 2013]. OMAERUV quantitative aerosol products have been evaluated by comparison to independent ground-based observations [*Torres et al.,* 2007; 2013; *Ahn et al., 2014; Jethva et al., 2014, Zhang et al., 2016*], airborne measurements [*Livingston et al.,* 2009] as well as to other satellite measurements [*Ahn et al., 2008; 2014; Gassó and Torres,* 2016].

Since early 2008, a reduction in OMI's spatial coverage associated with the onset of the so-called row anomaly has been observed. The row anomaly is believed to be the result of a physical obstruction that affects both Earth radiance and solar flux OMI measurements. Although in June 2007 only two of OMI's sixty viewing positions (or rows) were initially affected, the anomaly impact has extended to about 50% of the sensor's viewing positions. Because of the row-anomaly, the global daily coverage attainable during the first four years of operation is no longer possible. Worldwide-coverage is currently achieved in about two days.

The reduced viewing capability of the sensor may affect the accuracy of the long-term trends of OMI-derived products if the decrease in sampling frequency is so large that the diminished number of observing opportunities does not longer produce statistically equivalent spatial and temporal averages of the measured parameters. Because the OMI sensor still samples the same location every other day, no reduction on the statistical significance of the averaged retrieval results is expected when

using a subset of radiance observations over a reduced angular interval. Statistically non-equivalent results, however, may result if the angular distribution of the scattered and/or reflected incoming radiation is not realistically represented in the retrieval algorithm look-up tables. In version 1.4.2 of the OMAERUV algorithm, all aerosol models are assumed to be poly-dispersions of spherical particles [*Torres et al.,* 2007; 2013] and their scattering phase functions are calculated using Mie Theory. In the UVAI calculation, on the other hand, clouds are modelled as Lambertian opaque surfaces. Prompted by the loss of angular coverage associated with OMI's row anomaly, in this study we carry out a detailed examination of the effect of the reduction of OMI's angular sampling capability on the accuracy and statistical representativity of retrieved aerosol parameters.

In this paper, we investigate the effect of the reduced spatial coverage on the representativity of long-term OMI aerosol record, by examining the consistency of retrieval results using observation sub-sets associated with different scattering angle ranges. The full OMI viewing capability during the instrument's first three years of operation (2005-2007), allows a comparative analysis of time and space averaged aerosol parameters derived from different subsets of observations in scattering-angle-segregated retrievals that mimic the row anomaly effect. Section 2 describes in detail the row anomaly affecting the OMI performance followed by a discussion in section 3 of the methodology used in the analysis. Results of the across scan bias analysis conducted over different regions for the quantitative (AOD and SSA) and qualitative (UVAI) products are discussed in sections 4 and 5 respectively, including a discussion of changes in aerosol and cloud model representations required to address the issues identified in this study. Section 6 discusses the consequences of the row anomaly in the long-term record, followed by section 7 summarizing the results of the analysis.

## 2.0 Row Anomaly

Anomalous readings consisting on either increase or decrease of radiance signal at individual OMI viewing positions (or rows) started in early 2008. Because the initial manifestation of the problem was limited to individual rows, this instrumental issue has been referred to as *'row anomaly'*. Although the exact nature of the problem is not known, it is suspected that the row anomaly is the result of a physical obstruction that developed because of the loosening of fabric material covering the interior walls of the sensor. Detailed history of row anomaly onset and evolution is contained in lookup tables in the Level 1B software. The OMAERUV algorithm uses the row-anomaly detection method developed for the NASA OMI Total Ozone product (OMTO3) based on statistical analysis. It identifies total ozone anomalies in zonally averaged bands by comparison to data prior to the row anomaly onset [*Schenkeveld et al.,* 2017]. The row anomaly initially affecting two rows in June 2007 has extended to about 50% of the sensor's sixty rows as shown in Fig. 1.

The OMI row anomaly is not static as it slowly evolves over time at both long and short time-scales. It affects the quality of both level 1B spectral radiances and Level 2 products. The first sign of a row anomaly appeared on late June 2007, when a decrease in radiance signal affecting cross-track positions 54 and 55 was observed. Anomalous radiance readings affecting positions 38 through 43 at the northern section of the orbit were detected in May 2008. By December 2008, this effect propagated to position 45 along the entire orbit. Cross-track positions 28 through 45 show signs of degradation depending on

orbital position by late January 2009. Since then, the row anomaly has been varying more dynamically, affecting many rows, and occasionally releasing partial rows.

The rate of expansion of the row anomaly has slowed down since July 2011. The OMI's KNMI website (http://projects.knmi.nl/omi/research/product/rowanomaly-background.php ) provides a more complete description of this
instrumental issue.

### 3.0 Across Scan Bias Analysis

In this analysis, we have made use of the 2005-2007 OMAERUV aerosol record to calculate monthly mean values of
derived aerosol parameters over several regions. Viewing positions 1 through 30 on the West of nadir and positions 31 through 60 on the East are separately used. These two sets of observations are hereafter referred to as W and E to facilitate the discussion. Unless the decrease in sampling frequency when using only a subset of the total observations, leads to the missing of relevant aerosol events, the regional monthly means calculated using different observation sub-sets should be statistically equivalent. The retrieved aerosol parameters used in this study are AOD, SSA, and UVAI. Although the study was carried out
over several regions, for the sake of brevity, we report results at three regions: Northeast United States (NEUS); Southern Africa (SAF), and the Saharan Desert (SAH). The NEUS region (25N-45N, 60W-90W) is representative of areas predominantly associated with non-absorbing aerosols and clouds. SAF (5S-25S, 15E-35E) is a region known as an important source of carbonaceous aerosols, which are often observed mixed with clouds. In the SAH region (SAH, 16N-30N, 30E-10W), desert dust is the most abundant aerosol type, and cloud presence is significantly less than in the other two regions. Each
monthly mean W and E value of UVAI, AOD and SSA calculated for each region, is the average of tens of thousands of individual OMI level 2 observations.

Because the across-track push broom observing mode spans two different ranges of scattering angles, the occurrence of differences between the subsets of analysed data could be associated with the directionally inconsistent characterization of the angular  dependence of light scattering. The observed angular distribution is associated with the combined effect of the
scattering phase functions of Rayleigh scattering, and scattering by aerosol and cloud particles.  Unlike Rayleigh scattering, whose scattering phase function can be unambiguously calculated with great accuracy, the scattering phase functions of aerosols and clouds require detailed information on particle size, shape and optical properties of the scattering elements.

### 4.0 Effects on AOD and SSA retrievals

### 4.1 Analysis of results

Figure 2 shows monthly averaged values of retrieved AOD (top panel), and SSA (middle panel) treating observations East and West of the nadir separately over the SAF region. The red line depicts the calculated monthly averages using the E subset of observations while the blue line shows those obtained using only the W subset. The bottom panel of Fig. 2 shows the

temporal variability of the monthly averaged scattering angle for each side of the across-track segment. A 140°-165° scattering angle range prevails on the East side while a smaller range (117°-130°) is observed on the West branch of the scan.

The two sets of calculations yield similar results as a function of time, in spite of the different ranges of scattering angles corresponding to the two sets of observations. Largest AOD values and minimum SSA values occur in the August-October period when large amounts of absorbing carbonaceous aerosols are known to dominate the atmospheric aerosol load. The close agreement of the W and E retrieved products indicates an adequate representation of the actual aerosol scattering phase function. Similar results, not shown, were obtained over the NEUS region where sulphate and secondary organic aerosols are most common. The outcome of these comparative analyses suggests that the particle size distributions, spherical shape, and refractive index used in conjunction with Mie Theory for calculating the scattering phase functions of sulphate and carbonaceous aerosols in the OMAERUV algorithm [*Torres et al.,* 2007] adequately reproduce the observed scattering patterns of small spherical particles [*Kaufman et al.*, 1994; *Dubovik et al.,* 1998]. The calculated reflectances fields yield a consistent representation of the observed angular variability of OMI measurements in spite of the large difference in the scattering angle ranges.

Results of a similar comparison over the Saharan Desert region are shown in Figure 3. Unlike the close agreement between retrievals on both sides of the scan found over the SAF and NEUS regions, large across-track biases in retrieved AOD are observed over the Saharan region, particularly from February through September, when the atmospheric aerosol load is dominated by the presence of typically large dust particles. AOD retrieval differences (top panel) are significant during the February-September period, with largest discrepancies in April-July, which is the time of peak aerosol concentration over this region. On the other hand, during the months of minimum aerosol activity (October through January) AOD retrievals on both sides of nadir are in good agreement at the annual smallest values. The seasonality of the observed differences in SSA retrievals (middle panel) is almost diametrically opposed to that observed in the AOD case. Minima SSA retrieval differences between the two sets of observations are obtained during April through August, which are also the months of actual largest AOD as well as lowest SSA. Largest discrepancies between SSA retrievals on both sides of the across-track scan (about 0.02) are observed in the September-March period associated with both lowest aerosol load and highest SSA values. Average values of scattering angles for the West (110°-130°) and East (140°-165°) sides of nadir are shown on the bottom panel of Figure 3.

The scattering-angle-dependent results of retrieved aerosol optical depth and single scattering albedo over the Saharan Desert, suggests an inadequate model representation of the scattering properties of desert dust particles assumed to be spherically shaped in the OMAERUV algorithm. Desert dust aerosols are known to be irregularly shaped, large particles whose phase function may deviate significantly from that of a spherical model at scattering angles larger than about 80° for non-absorbing particles. The role of particle shape assumption in the retrieval of desert dust properties was recently identified as an important source of uncertainty in the OMAERUV algorithm [*Gassó and Torres,* 2016]. An analysis of the uncertainty in retrieved AOD and SSA associated with the spherical shape assumption of desert dust particles is presented next.

**4.2 Sensitivity Analysis**

Accurate representation of the scattering phase function of large nonspherical particles requires the use of adequate analytical tools such as T-matrix theory [*Waterman,* 1971; *Bohren and Singham*, 1991; *Mishchenko and Travis,* 1994], geometric optics [*Yang and Liou,* 1996] or a combination of the two approaches [*Dubovik et al,* 2006]. The simplest non-spherical shapes for which exact analytical solutions can be obtained are spheroids whose shapes are characterized by their aspect or axis ratio ($\varepsilon$). Although it has been shown that the spheroid assumption using T-matrix theory reproduces laboratory measurements of the scattering phase function significantly better than spheres, it breaks down for highly elongated and flattened spheroids of very large size parameter ($2\pi r/\lambda$) [*Mishchenko et al.,* 2002]. The Geometric Optics method documented by *Yang and Liou* [2000] accurately accounts for scattering effects of spheroids of aspects ratios and size parameters beyond T-matrix capabilities. *Dubovik et al.* [2006] combined the advantages of the T-matrix and Geometric Optics to create a set of look-up tables of phase matrix elements at a 1° scattering angle resolution. As explained in *Dubovik et al.* [2006], phase matrix elements were calculated for real refractive index values between 1.33 and 1.6; imaginary refractive index between 0.0005 and 0.5; aspects ratios in the range 0.3 to 3.0; and size parameters from 0.012 to 625. In the following analysis, we used the *Dubovik et al.* [2006] kernels and the associated software package available from the author to extract the phase matrix elements associated with the particle size distribution and refractive index of the standard OMAERUV desert dust models [*Torres* et al., 2007].

Retrieval errors associated with aerosol particles non-sphericity in the near UV were first analysed in the context of volcanic ash detection and characterization [*Krotkov et al.,* 1999]. The effect of nonspherical particles in the interpretation of satellite near UV measurements in the presence of desert dust aerosol particles was addressed by *Syniuk et al.* [2003] and *Gassó and Torres* [2016]. In this section, we carry out a sensitivity analysis to evaluate the effect of neglecting aerosol non-sphericity in the retrieval of AOD and SSA of desert dust aerosols by the OMAERUV algorithm.

A known difficulty in the treatment of nonspherical particles is the need of prescribing the fraction of nonspherical elements in the polydispersion, as well as making assumptions on the prevailing aspect ratio values. To address those issues, we follow the statistically optimized approach of *Dubovik et al.* [2011] to account for mixtures of spherical and nonspherical particles, as well as mixtures of spheroids of varying $\varepsilon$ values as suggested earlier by *Mishchenko et al.,* [1997]. In *Dubovik et al.* [2006, 2011], the aerosol polydispersion is modelled as a mixture of randomly oriented spheroids. Each size bin consists of a size independent distribution of $\varepsilon$ ranging from 0.33 to 2.98 which includes flattened oblate spheroids ($\varepsilon<1$), elongated prolate spheroids ($\varepsilon>1$), in addition to spheres ($\varepsilon=1$). The aspect ratio is distributed in 25 bins, with each $\varepsilon$ bin having a fixed weight such that the sum of all weights equals unity as shown in Figure 4. This modelling approach, that closely reproduces the laboratory measured single scattering matrices of mineral dust (Feldspar) reported by *Volten et al* [2001], is currently applied in the operational AERONET (AErosol RObotic NETwork) inversion of measured sky radiances [*Dubovik et al.,* 2006]. The resulting spheroid scattering phase function and it sphere-equivalent representation at 388 nm for single scattering albedo 0.9 are shown in Figure 5. Additional calculations (not shown) as a function of aerosol absorption, indicate that in the near UV, the observed sphere-spheroid phase function difference in the 80°-150° scattering angle range is largest for non-absorbing aerosols, and reduces significantly for SSA values 0.82 and lower.

Scattering matrix elements extracted from the *Dubovik et al.* [2006] kernel package were fed to a vector radiative transfer code to generate top-of-the-atmosphere (TOA) radiances at 354 and 388 nm for an atmosphere containing the polydispersion of nonspherical particles discussed above. The calculated radiances associated with specific AOD and SSA values were used as input to a research version of the OMAERUV algorithm that assumes desert dust to be spherical particles.

Figure 6 (top) illustrates simulated AOD retrieval errors (in percent) as a function of scattering angle when nonspherical aerosol models of varying single scattering albedo are treated as spherical  particles in the inversion procedure.  The vertical lines indicate the range of average scattering angle associated with the West (red lines) and East (blue lines) sections of the across-track segment as shown on the bottom panel of Figure 3. AOD retrieval errors transition from overestimations to underestimations at about 155° scattering angle.

On the 110°-130 scattering angle range, associated with OMI's average viewing geometry on the West side of the scan, AOD errors are always positive, and show little angular dependence. Retrieval errors vary between about 13% for the model of lowest SSA value used in the analysis (0.83) and close to 23% for the least absorbing case (SSA = 0.97).  For the range of average scattering angles (140°-165°) corresponding to the East side of OMI's scan, positive AOD retrieval errors decrease rapidly with scattering angle. Much larger underestimation errors take place for scattering angles larger than 155°, with the

error rapidly increasing with scattering angle.  For weakly absorbing aerosols, errors in excess of 100% are possible at scattering angles larger than 165°. Because the inversion algorithm simultaneously retrieves both AOD and SSA, no AOD is retrieved in cases when the retrieved SSA is unphysical as discussed next.

Absolute errors in retrieved SSA associated with the spherical shape assumption for the same set of aerosol models as in the previous discussion are shown on the bottom panel of Fig. 6. For scattering angles smaller than about 155°, small negative

and positive errors (absolute value less than 0.01) occur.  For aerosols of actual SSA larger than about 0.92, the retrieved value is slightly overestimated whereas for more absorbing particles, a small SSA underestimation takes place. The error increases with decreasing SSA and is largest (about -0.01) for the aerosol model of lowest modelled SSA (0.83).

At scattering angles larger than 155°, SSA retrievals are underestimated for moderately absorbing aerosols (SSA smaller than 0.9) yielding errors as large as -0.02 at 165° scattering angle, and even larger in the near backscatter direction. For weakly

absorbing and non-absorbing aerosols, small positive retrieval errors are obtained.

The previous sensitivity analysis allows the interpretation of the observed W-E differences in retrieved AOD and SSA depicted in Figure 3. According to these results, retrieved AOD values on the West side of the scan, where the average scattering angle stays in the range 110°-130°, are overestimated whereas those on the East side are underestimated. Because the average scattering angle on the East branch of the scan remains above 160° from March through September, the resulting

AOD is underestimated. It can also be inferred that the magnitude of the underestimation is much larger than that of the overestimation that explains the large difference in W-E dust AOD retrievals in Figure 3. Regarding the SSA retrieval, the observed W-E differences (as large as 0.02 but within 0.01 most of the time) in retrieved SSA are consistent with the results of the sensitivity analysis that indicates that, in addition to the angular dependence, the magnitude of the retrieval error also depends on the actual SSA value.

**4.3 Application to OMI observations**

A new set of look-up tables at 354 and 388 nm calculated for the spheroidal particle shape model described in section 4.2 were used in a research version of the OMAERUV inversion algorithm. Retrieval results are shown in Figure 7. The resulting W-E AOD across-track bias was reduced from values as high as 0.25 in June 2006 (Fig. 3) to about 0.02 (Fig.7). The amplitude of the across-track SSA bias (middle panel) was reduced from about 0.025 to 0.015. However, a small bias of about 0.007 is still present. Because of cancellation of small remaining biases in AOD and SSA, the calculated AAOD W-E differences are smaller than 0.005 (bottom panel).

As shown in Fig. 7, accounting for the non-sphericity of aerosol particles virtually eliminates the large across-scan AOD bias observed when the spherical particle shape approximation is used. The non-spherical shape approximation reduces further the small error SSA error associated with the spherical particle shape assumption

**5.0 UV Aerosol Index**

**5.1 Scan bias Analysis**

The UVAI calculated as shown in Appendix A, is a residual parameter that quantifies the difference between measured and calculated ratios of UV radiances [*Torres et al.,* 1998]. When all radiative transfer processes are effectively accounted for, the UVAI should, by definition, be zero. Unlike with the retrieval of quantitative aerosol parameters that require cloud screening, the UVAI is calculated for all observations regardless of cloudiness conditions or ice/snow presence.

In this section, we examine the directional consistency of the UVAI parameter in a manner similar to that in the previous section. Figure 9 (top panel) depicts monthly mean UVAI over the NEUS region calculated separately for the West and East sections of the scan. Near-zero W-E differences are observed in February followed by a rapid increase to a maximum of about 0.4 from May through July when it starts to decrease again to near-zero difference in October. A similar situation, observed over the SAF region, is depicted in the middle panel of Figure 9. Here, the absolute W-E differences are largest in January (about 0.5), decrease to zero by May and remain low until August, when the two average values start diverging again. The time series of UVAI is also shown for the SAH region in the bottom panel of Figure 9. Largest W-E differences of about 0.2 UVAI units are observed during the spring months.

There is a clear contrast between the small W-E differences in the SAH region and the much larger ones over the other two regions in Fig.9. The most relevant difference between the SAH region and the NEUS and SAF regions is the significantly reduced levels of cloudiness in the former. Thus, the much smaller UVAI across-scan bias over the SAH region where cloudiness is generally very low during the entire year, and the observed season dependent angular bias over the NEUS and SAF regions, suggest that the angular dependence of cloud scattering effects may not have been adequately accounted for in the UVAI calculation. A more detailed visualization of the cross-track angular dependence of the UVAI over the NEUS region in January and July 2005 is shown in Figure10.

The above analysis was also carried out using the Modified LER UVAI definition described in Appendix A (not shown here for the sake of brevity). As with the SLER UVAI definition, a clear, although slightly reduced in magnitude, across track bias effect was observed. In the next section, we will examine if the observed W-E differences reported here can be explained by the way cloud reflection effects are treated in the UVAI computation.

## 5.2 Parameterization of cloud effects in UVAI calculation

The UVAI in the OMAERUV algorithm is currently calculated using the SLER approximation described in Appendix A. In this approach the combined effect of surface and cloud reflection, as well as scattering and absorption aerosol effects is represented by an opaque Lambertian reflector located at the surface. In the presence of clouds, therefore, the SLER representation does not capture the angular variability associated with the scattering phase function of clouds. Such an approximation may be, therefore, responsible for the across-scan UVAI bias observed by the OMI sensor as discussed in section 4. To test that hypothesis, we have developed an alternate way of calculating the UVAI that explicitly accounts for scattering effects of water clouds following the results of a previous study showing that the use of Mie scattering theory reproduces remarkably well the satellite observed field of backscattered UV radiation in a cloudy atmosphere [*Ahmad et al.,* 2004]. In this approach, it is assumed that the radiance measured by the sensor at pixel level emanates from a combination of clear and cloudy conditions ( $I_\lambda^s$ and $I_\lambda^C$ ) involving a cloud of fixed optical depth and varying cloud fraction. The $I_\lambda^s$ terms are calculated using wavelength dependent climatological values of surface albedo, derived from analysis of the 10-year long-term OMI record of minimum reflectivity. The $I_\lambda^C$ terms, on the other hand, are calculated using Mie scattering theory for an assumed water cloud model characterized by a Modified Gamma particle size distribution associated with cumulus clouds [*Deirmendjian,* 1964] and commonly referred to as the C1 model [*Deirmendjian,* 1969]. The wavelength-dependent refractive index in the near UV was taken from the *Hale and Querry* [19731973] data base. Although the refractive index difference between the two UV channels is only 0.003, not accounting for it introduces an angle dependent UVAI signal as large as 0.3. Calculations were carried out at prescribed cloud top and bottom levels, and fixed cloud optical depth (COD). The choice of COD value (10) was based on the highest frequency of occurrence of this value reported by MODIS observations [*King et al.,* 2013]. A wavelength independent effective cloud fraction, $f_g$, is calculated from equation

$$f_g = \frac{I_{\lambda_0}^{obs} - I_{\lambda_0}^s}{I_{\lambda_0}^C - I_{\lambda_0}^s} \qquad (1).$$

When the resulting cloud fraction is larger than unity, overcast sky conditions are assumed (i.e., $f_g=1.0$), and a new $I_\lambda^C$ term for COD value larger than 10 that matches $I_{\lambda_0}^{obs}$ is derived. $I_\lambda^{cal}$ values are then obtained from equation,

$$I_\lambda^{cal} = (1.0 - f_g)I_\lambda^s + f I_\lambda^C \quad (2),$$

and used as input in the calculation of the UVAI in equation A-1.

Although this way of calculating the $I_\lambda^{cal}$ term is conceptually similar to the MLER method [*McPeters et al, 1998; Penning De Vries and Wagner*, 2011] described in Appendix A, there are important differences that affect the magnitude of the calculated values. The major difference is that while in the MLER approach clouds are modelled as Lambertian opaque surfaces using Rayleigh scattering calculations, the method tested here treats clouds as poly-dispersions of liquid water droplets, and uses Mie radiative transfer theory in the forward calculations. The calculated value is sensitive to the choice of COD for which a value of 10 has been assumed in this work.  Except at high solar zenith conditions, the calculations are insensitive to assumed cloud top and bottom levels. Accounting for the spectral dependence of surface albedo is also an important difference that will affect the magnitude of the calculated radiances and resulting UVAI values. For the COD value used here the resulting Mie-UVAI is generally 0.3 larger than the SLER definition. This difference increases with assumed COD.

**5.3 Evaluation of Results**

Figure 11 depicts the resulting cross-track angular dependence of the UVAI over the NEUS region in January and July 2005 when applying the Mie-based approach. For both months, the resulting angular dependence is significantly reduced in relation to that shown in Figure 10 when using the SLER-based method of UVAI calculation. Largest improvements are observed for positions 1 through 20 associated with scattering angle ranges 80-110 for January and 100-140 for July. Overall, the Mie UVAI is closer to zero at all angular positions yielding very small W-E differences as shown in Fig 12.  The better performance of the Mie-based approach is consistent with the results of *Ahmad et al.* [2004] who found out that Mie radiative transfer calculations using the C1 cloud model reproduced satellite observed UV radiances over a large range of viewing conditions better than parameterizations that model clouds as opaque surfaces.

Figure 12 shows the comparison of resulting across-scan bias in UVAI calculated using the Mie-based definition tested here. The W-E biases observed using the standard SLER UVAI definition application in Figure 9, have been reduced in the three regions used in the analysis. Significant cross-track bias reduction is observed over the NEUS and SAF regions where high cloudiness levels prevail.

**6.0 Discussion**

The improved radiative transfer modelling in the presence of desert dust aerosols and water clouds, as discussed in this paper, was incorporated in a revised version of the OMAERUV algorithm. The effect of these upgrades on the global product, and the impact of the row anomaly on the long-term OMAERUV regional records are briefly discussed here. A more detailed discussion of the global long-term OMAERUV record will be given in a forthcoming publication.

A global comparison of resulting Mie and SLER based UVAI definitions under cloudy conditions are shown in Fig. 12 for August 20, 2007. The across-scan bias of the Mie-based UVAI map, shown on the middle panel, is significantly reduced

in relation to the corresponding SLER-based UVAI map depicted in the top panel. To facilitate the comparison, the bottom panel of Fig 12 shows the associated reflectivity field.  A clear reduction in across-scan Mie UVAI bias is observed over cloudy regions of reflectivity larger than about 20%.

      Figure 13 (upper panel) shows the W-E differences between the monthly averages of UVAI obtained with the SLER (blue line) and Mie-based (red line) calculation approaches over the SAF region for the 2005-2014 period. Both results show repeatable seasonal cycles over the first four years of observations. The SLER UVAI across-track differences oscillate between about -0.6 in winter to about 0 in winter whereas the Mie UVAI W-E differences shows much less seasonal variability (-0.1 to 0.2). An overall drop of the W-E UVAI differences is apparent by the summer of 2009 when the row anomaly has fully developed. A decrease of about 0.2 is observed in the SLER UVAI data whereas a smaller change (~0.1) can be seen in the Mie-based UVAI definition.  From then on, the winter W-E UVAI difference stabilizes at -0.8 for the SLER method, and at -0.2 for the Mie-based definition. The summer maxima, on the other hand, slowly increases with time after 2010 for both parameters but a more rapid increase is apparent for the SLER UVAI term. Overall, the effect of the row anomaly is larger and more noticeable in the SLER UVAI definition. Similar results (not shown) are also observed over the NEUS region. The eleven-year record of monthly average UVAI over the SAF region obtained by both definitions is shown in the lower panel of Figure 13. Overall, the Mie-based UVAI record is about 0.3 higher. This increase moves up the minima UVAI values associated with water clouds from negative (~ -0.3) in the SLER definition to nearly zero in the Mie-based calculation. The rapid decrease in the SLER UVAI annual minimum in early 2008, following the onset of the row anomaly, is not present in the Mie-based UVAI record. Both records show an increasing trend beginning in 2009. The SLER UVAI shows a 0.04/yr. increase, whereas the Mie-based UVAI is about half that value.

      Upper panel of Figure 14 depicts the eleven-year temporal record of the across-track AOD bias for both spherical and spheroidal model representations of desert dust aerosols. The W-E differences for the spheroidal model (red line) oscillate around 0.05 but are never larger than 0.1, with no clearly observable effect of the onset of the row anomaly. The spherical model (blue line), on the other hand, produces across-track differences as large as 0.25 during the first two years, and a decrease to 0.15 coincident with the initial loss of viewing capability associated with the row anomaly. Following the row anomaly onset, the spherical model yields seasonal and inter-annual variability of across-track AOD differences as high as 0.2 towards the end of the shown temporal record. The resulting multi-annual records of AOD over the SAH region as calculated by the spherical and spheroidal models are shown on the bottom panel of Figure 14. During the first four years of the record, the spherical model underestimates spring and summer monthly averages by about 0.05 with respect to the spheroidal model. Starting in 2009, after the full development of the row anomaly, the bias goes down, and the spherical model results are closer to those of the spheroidal model. The reason for this bias reduction is the exclusion of observations from most of the rows in the East across-track segment of the orbits (yielding larger errors associated with the erroneous phase function assumption) as they are affected by the row anomaly.

**7.0 Summary and conclusions**

        The OMI sensor is currently affected by a loss of spatial coverage commonly referred to as row-anomaly. Although an unequivocal explanation of the row anomaly does not exist, it is believed to be the result of an internal obstruction affecting the sensor's across-track viewing capability. Prompted by the loss of angular coverage associated with OMI's row anomaly, we carried out a detailed examination of the effect of the reduction of OMI's angular sampling capability on the accuracy and statistical representativity of retrieved aerosol parameters, and the consequences of the row anomaly on the long-term record derived from OMI observations.

        An analysis of the effect of OMI's reduced viewing capability on the representativity of time and space averaged aerosol products has been carried out. Regional monthly average values of AOD, SSA and UVAI, retrieved from observations using two different sets of scattering angles associated with off-nadir viewing geometries on the East and on the West of nadir were compared. Close agreement between the two sets of averaged retrieval results was expected under the assumption that the forward-model-calculated angular dependence of the radiation field in the presence of aerosols and clouds realistically reproduces the actual scattering patterns of cloud and aerosol particles.

        The expected agreement in the retrieved AOD and SSA properties was obtained in cases when the aerosol load consisted of either carbonaceous or sulphate aerosols, known to be predominantly small spherical particles, and whose scattering phase function is calculated using Mie scattering theory. Large differences in retrieved AOD values, however, were found for retrieval conditions when the aerosol load was mainly desert dust, made up of predominantly large non-spherical particles. Small errors in retrieved dust SSA were also found. Because the retrieval algorithm also makes use of Mie Theory to calculate the scattering properties of desert dust aerosols, the wrong particle shape assumption appeared to be the likely source of the discrepancy. The analysis was then repeated using a research version of the retrieval algorithm in which the scattering properties of desert dust aerosols were obtained using a combination of T-matrix and geometric optics calculations. The observed differences between the two retrievals under the spherical shape assumption reduced significantly when using the non-spherical particle shape approximation.

        Differences between the two sets of observations were also observed in the retrieved UVAI over the NEUS and SAF regions where clouds are persistently present. No significant differences were observed under the predominantly cloud free conditions prevailing over the SAH region. The source of the difference turned out to be the currently used parameterization of cloud scattering effects in the calculation of UVAI in which clouds are treated as opaque Lambertian reflectors located at the surface. In the presence of clouds, this representation does not capture the angular variability associated with the scattering effect of clouds. A modified version of the UVAI algorithm in which a more physically based approach is used to incorporate cloud scattering of incoming radiation was developed and tested. Obtained results largely eliminated the observed differences between the two sets of observations.

        The analysis discussed here has uncovered important algorithmic deficiencies associated with the model representation of the angular dependence of scattering effects of desert dust aerosols and cloud droplets. In addition to the

documented east-west asymmetries in the magnitude of the retrieved aerosol parameters, the use of inaccurate scattering phase functions of desert dust aerosols and clouds introduces spurious features in the long-term record when a range of scattering angles is no longer available because of the row anomaly. The more accurate representation of the scattering patterns of water clouds, and the spheroidal particle shape assumption in the retrieval of desert dust properties eliminates the observed inconsistency of aerosol retrieval results associated with the separate use of east-of-nadir or west-of nadir observations. The recommended improvements in the modelling of scattering by cloud droplets and desert dust aerosols also reduces the magnitude of row-anomaly related discontinuities in the long-term record of the OMI aerosol products.

## Acknowledgements

This work was carried out under the auspices of a NASA Aura ROSES proposal, NNH13ZDA001N-AURA, Ken Jucks (program manager).

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

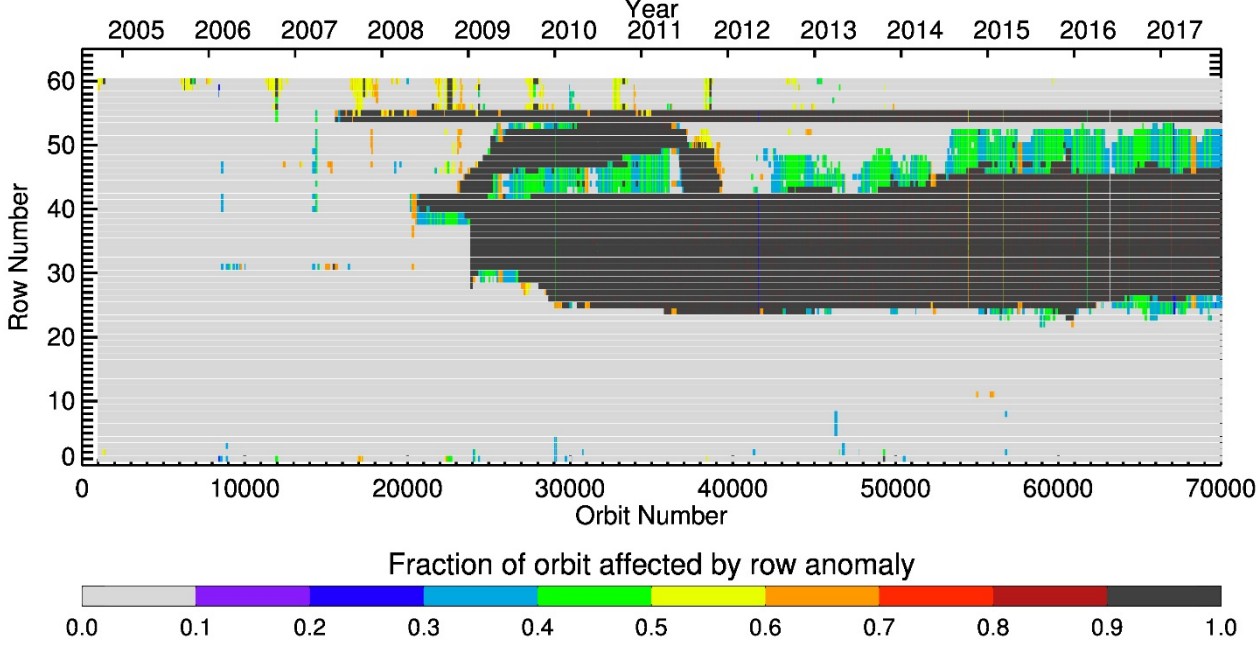

**Figure 1. Temporal evolution of the row anomaly as a function of orbit number (bottom) and year (top). The color scale indicates the fraction of orbit impacted by the anomaly.**

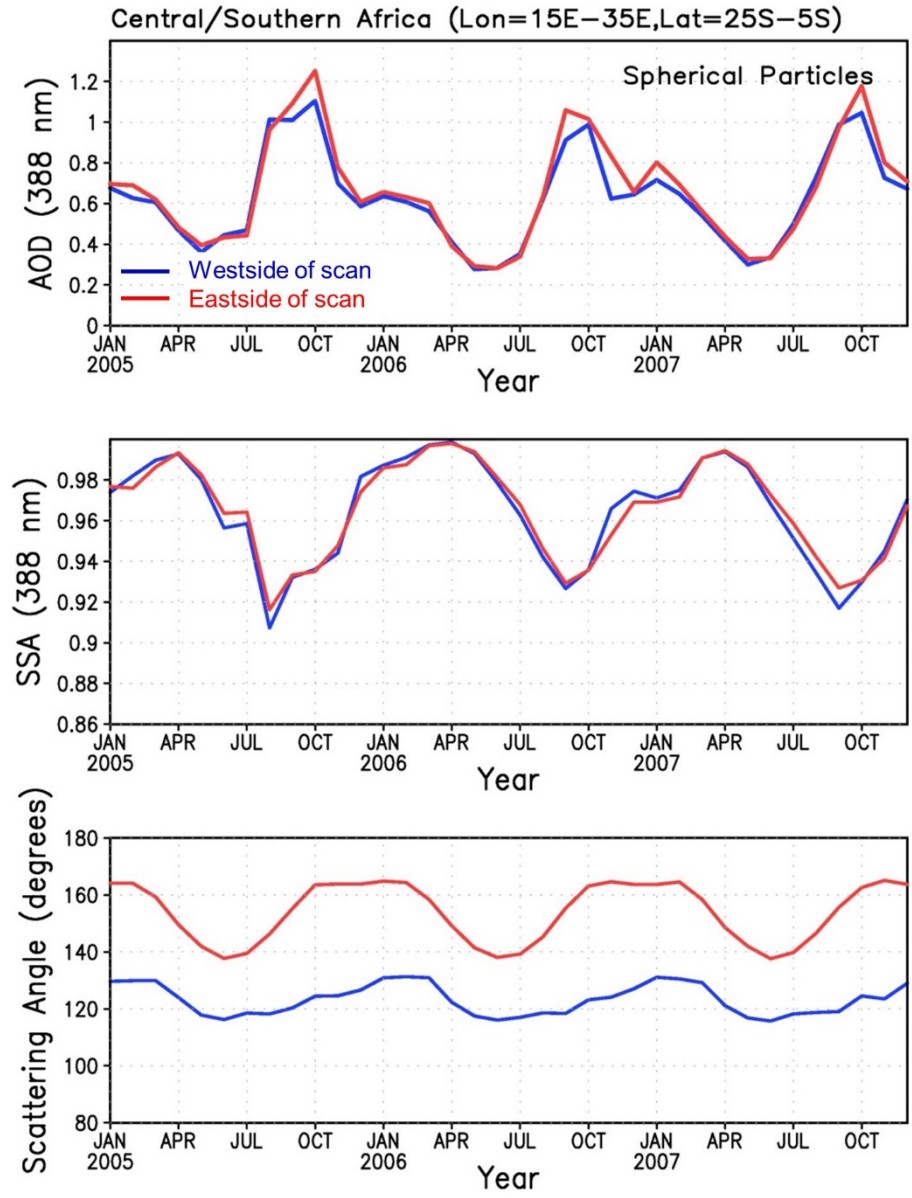

**Figure 2.** Time series of monthly averages of OMI retrieved AOD (top panel) and SSA (middle panel) over Southern Africa. The red line represents the average calculated from observations by rows 1 through 30 whereas the blue line indicate the resulting averages when using rows 31 through 60. The bottom panel shows the average scattering angle for the two data subsets. See text for details.

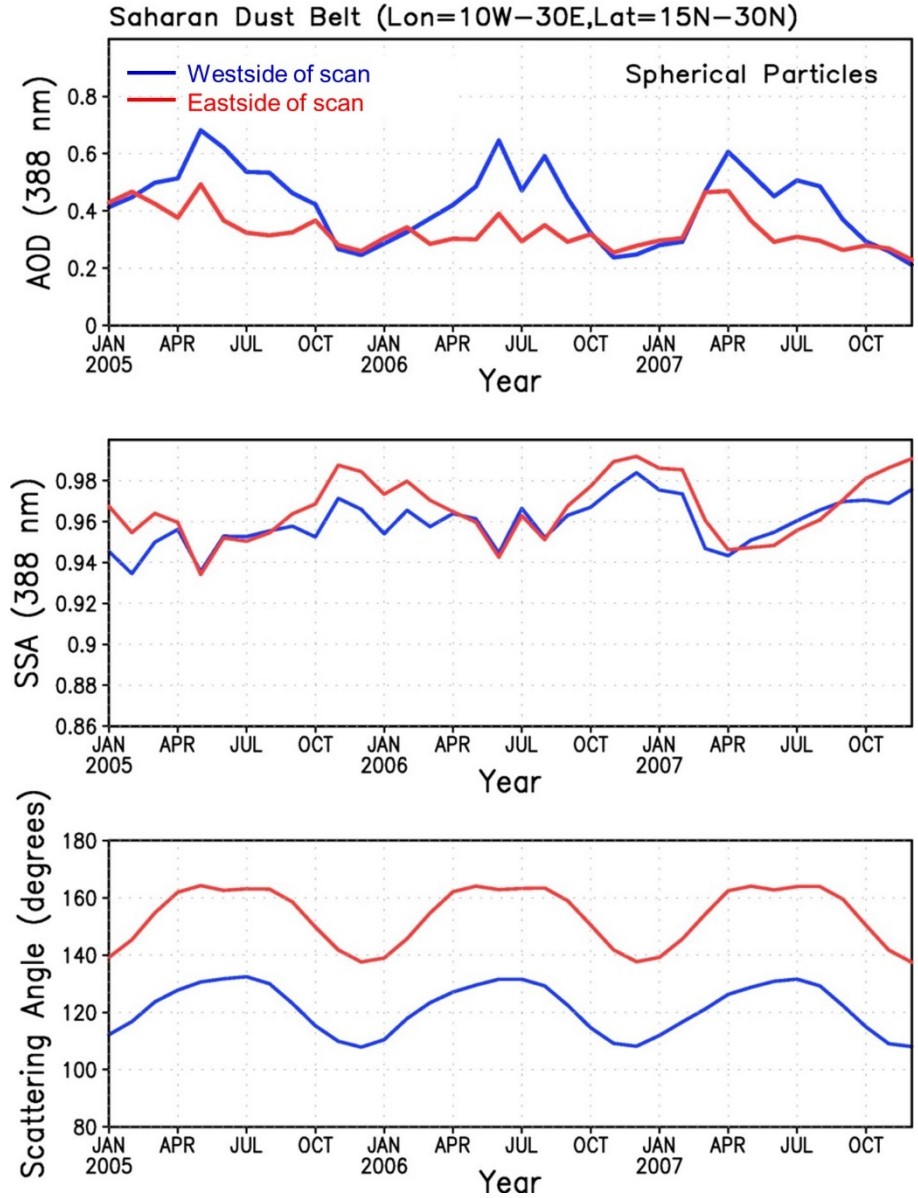

**Figure 3. As in Fig. 2 for the Saharan desert region.**

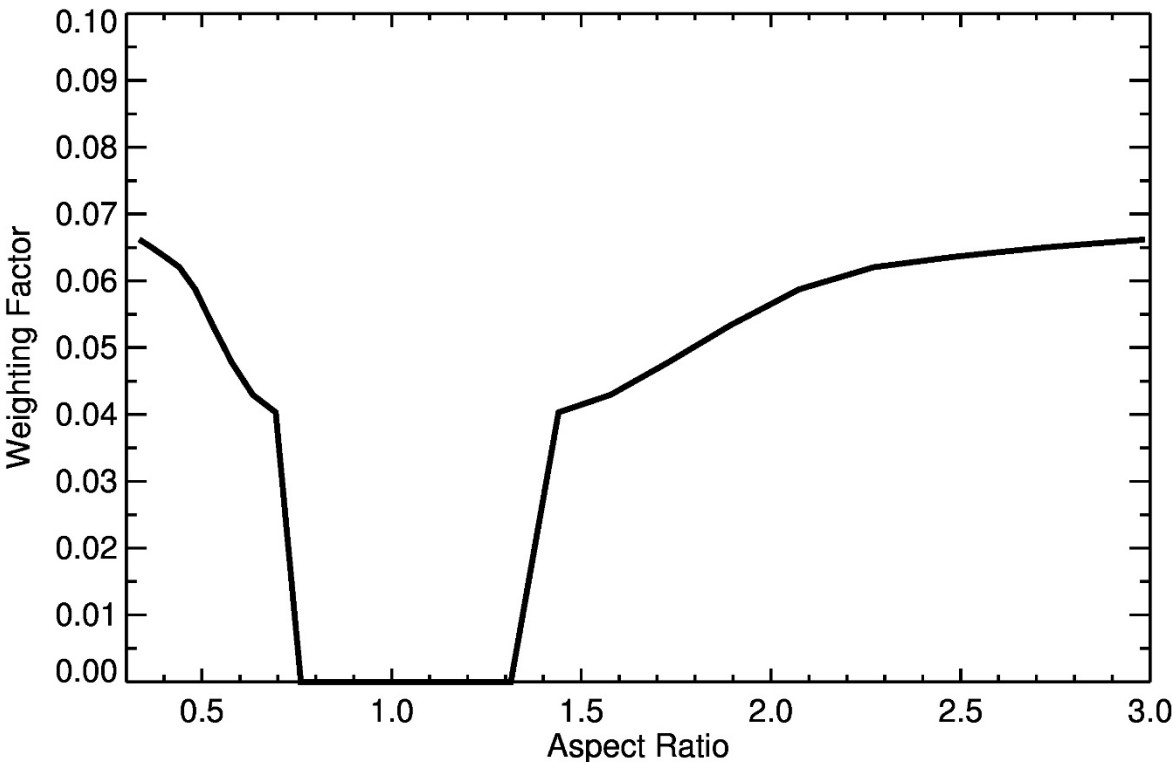

Figure 4. Aspect ratio (ε) weighted distribution for aerosol spheroid polydispersion.

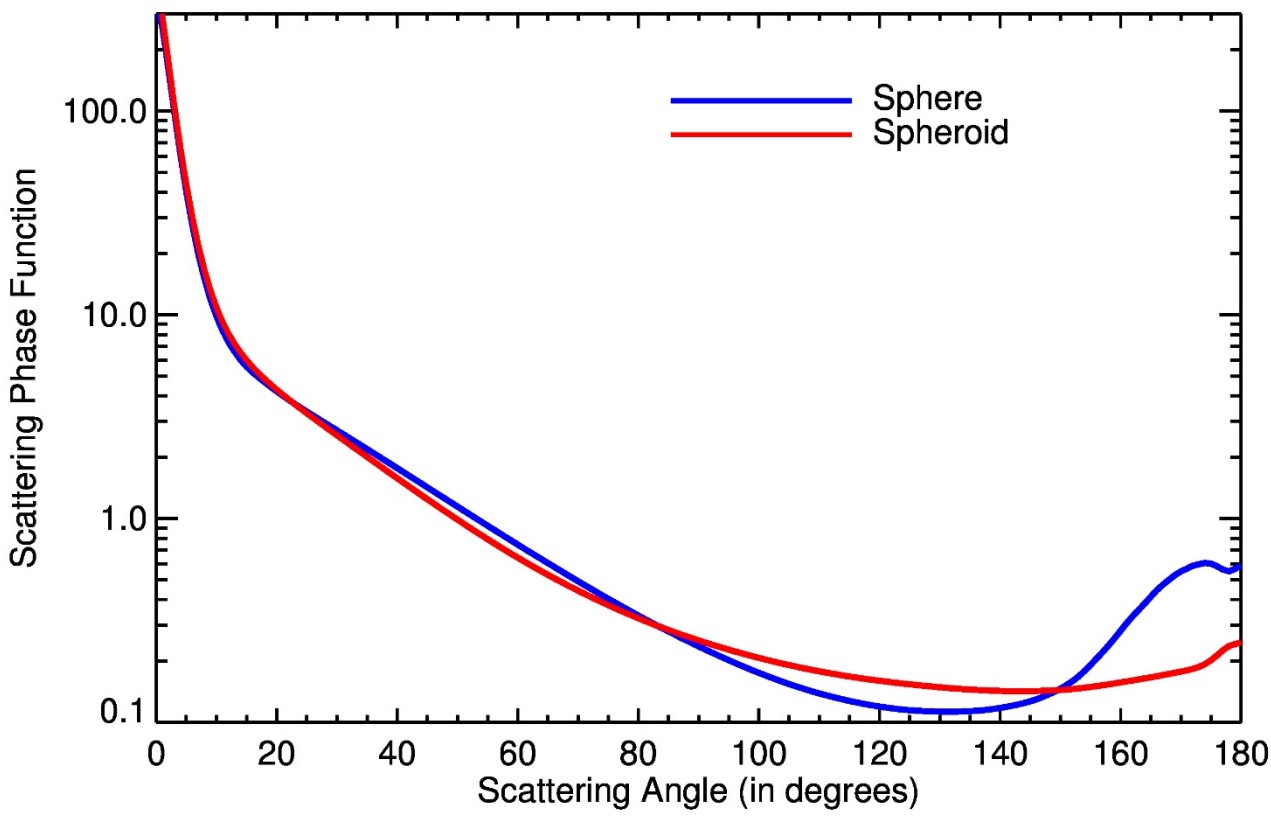

Fig.5 Calculated scattering phase function of spheres (blue) and spheroids (red) at 388 nm for SSA=0.9. Assumed refractive index 1.55 + 0.00405i. See text for additional details.

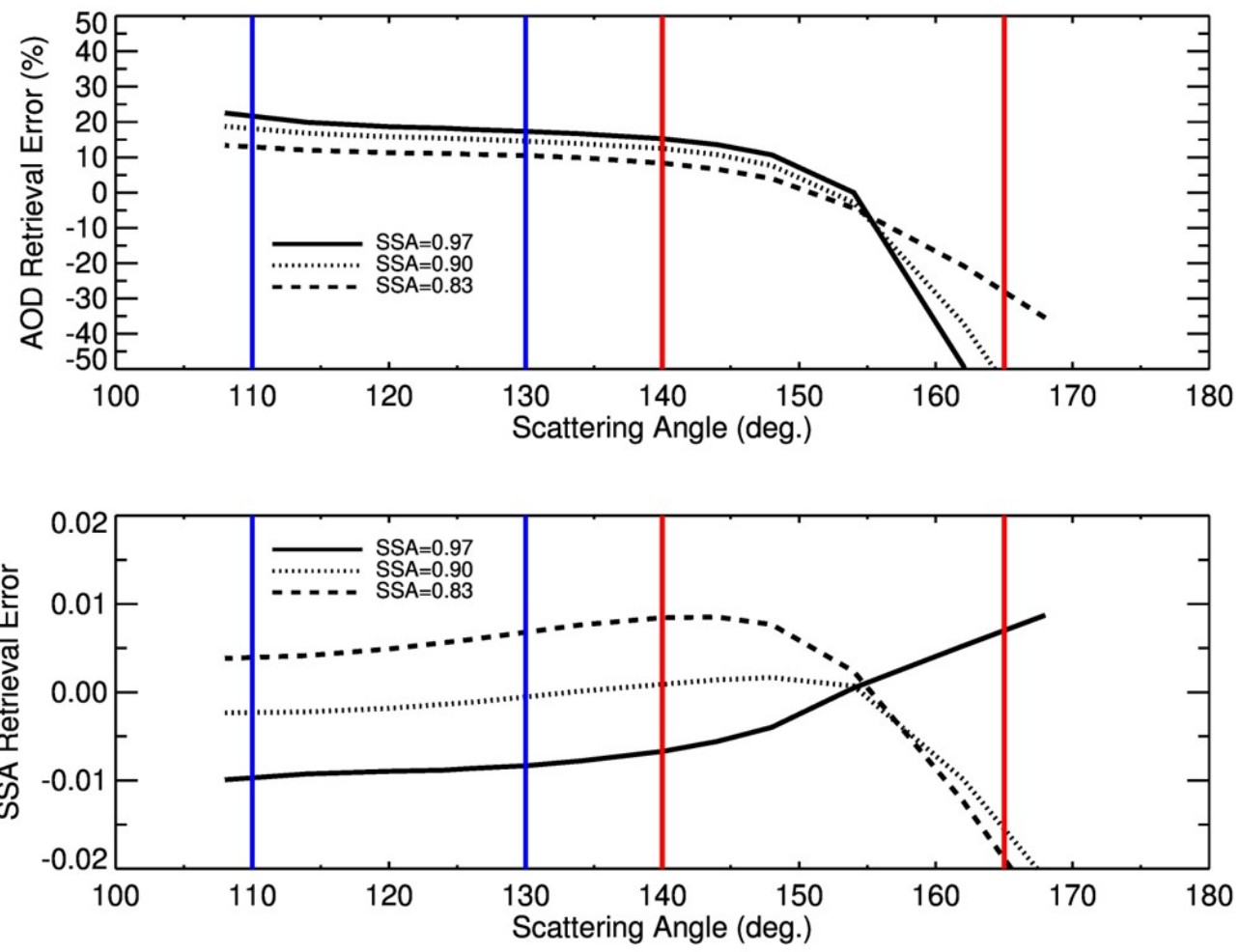

**Figure 6.** Percent error in retrieved AOD (top) for a non-spherical aerosol polydispersion of optical depth 1.0 (500 nm) assumed to be spherical in the retrieval process. Calculations were done for different SSA values: 0.97 (solid line), 0.90 (dotted line), and 0.83 (dashed line). Vertical lines indicate the range of scattering angles in Fig. 3 for rows 1-30 (blue) and rows 31-60 (red). The bottom panel shows the resulting absolute error in retrieved SSA associated with the spherical particle assumption. See text for details.

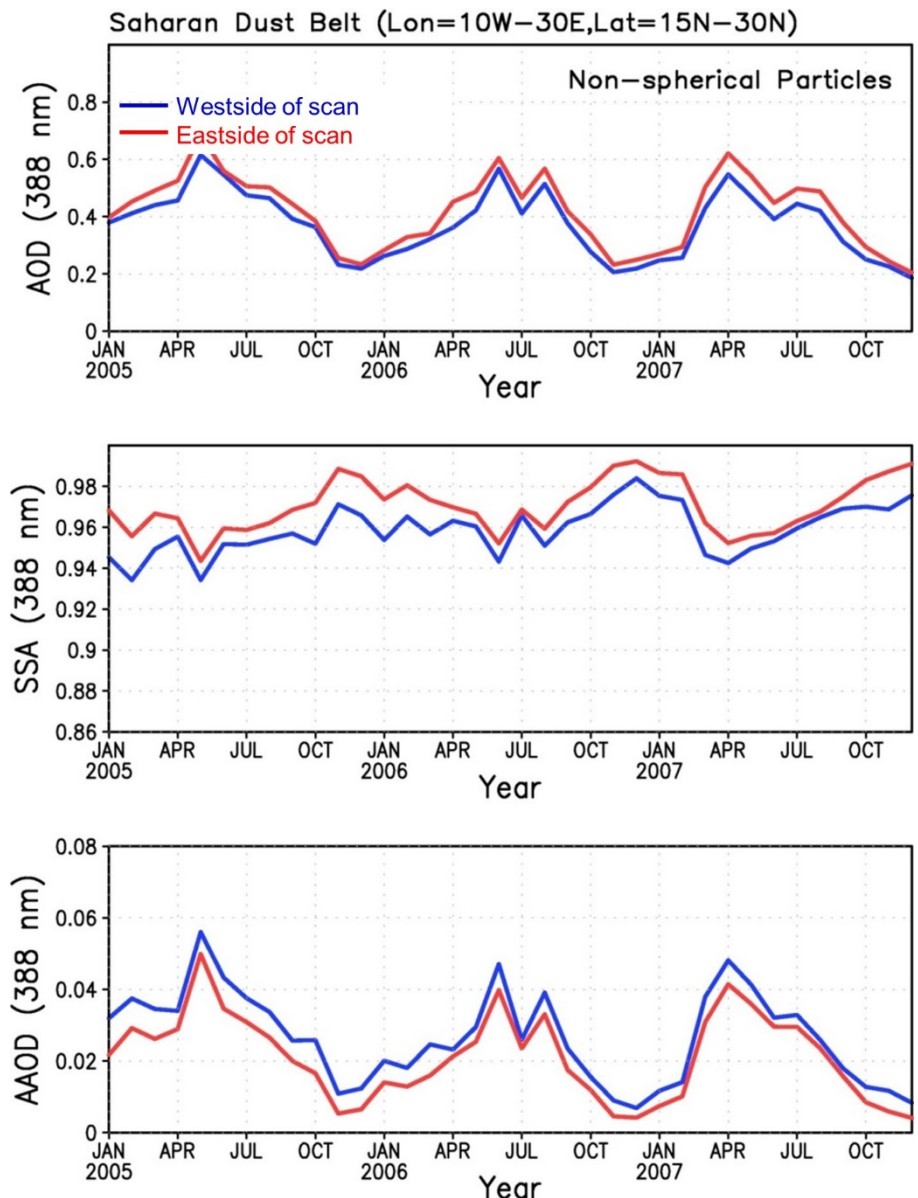

**Figure7. Time series of monthly averages of retrieved AOD (top panel), SSA (middle panel), and AAOD (bottom panel) over the Saharan Desert. Retrievals carried out using an OMI research algorithm that accounts for aerosol non-sphericity. Line color interpretation same as in Figure 3.**

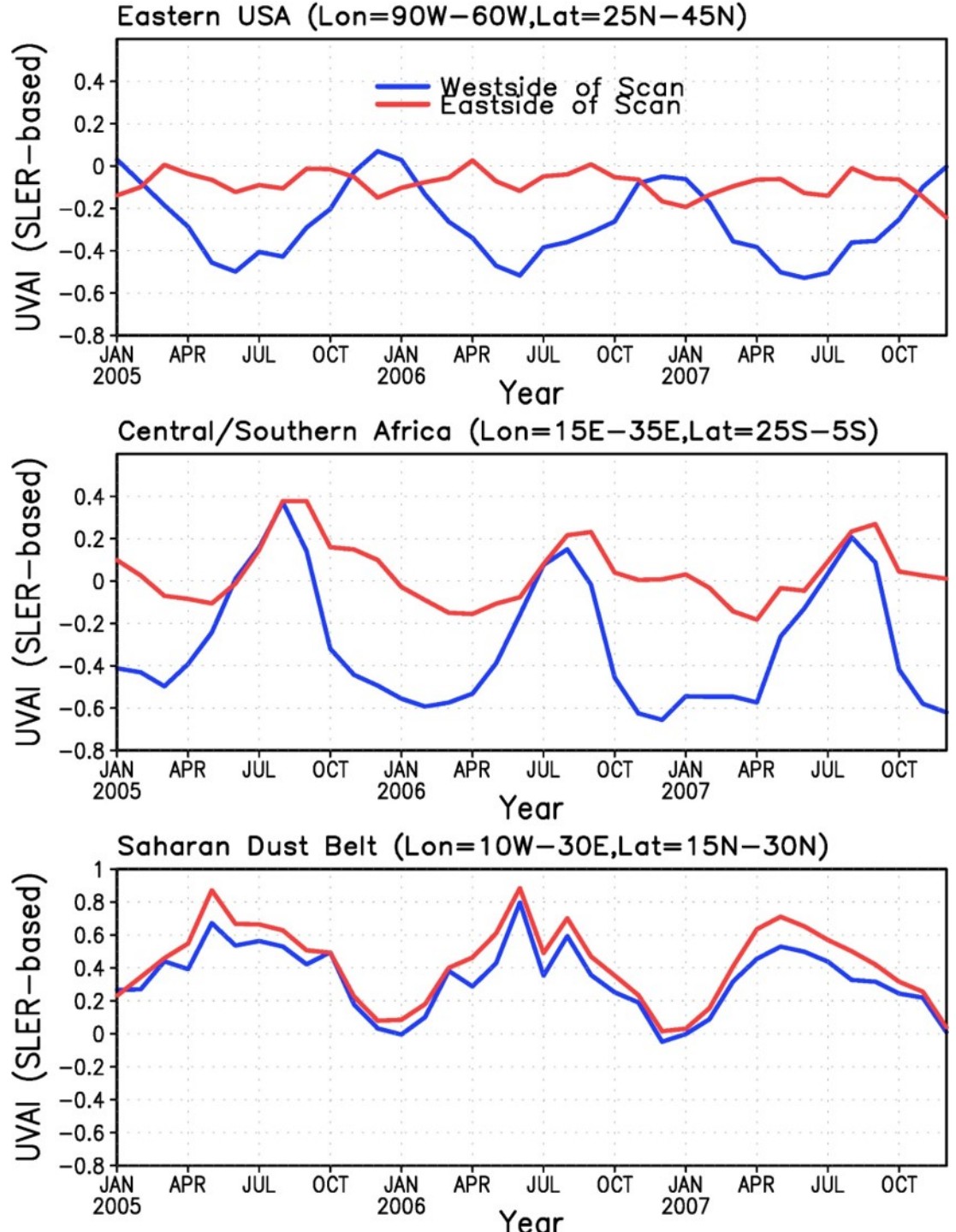

**Figure8. Time series of calculated monthly average UV-AI over the Northeast US (top), SAF (middle), and SAH (bottom) regions using rows 1 to 30 (blue) and 31 to 60 (red).**

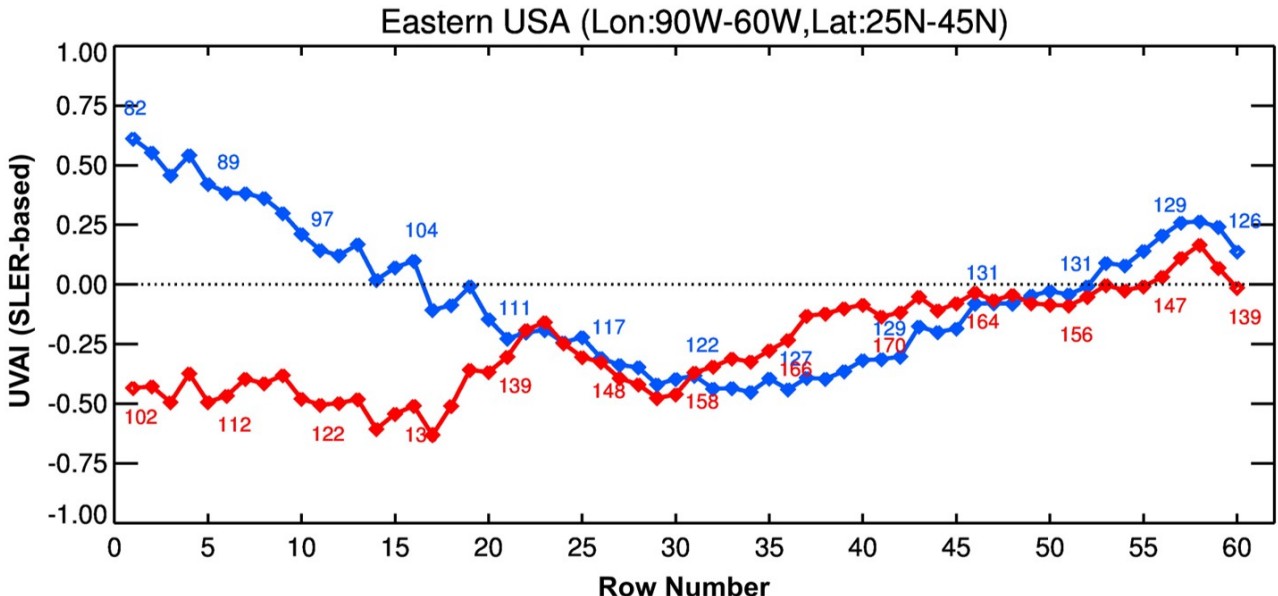

**Figure9.** Monthly average values of retrieved LER-based UVAI as a function of viewing position (or row) number. Results for January (blue) and July (red) 2005 are shown. Numbers indicate the monthly average scattering angle for each row.

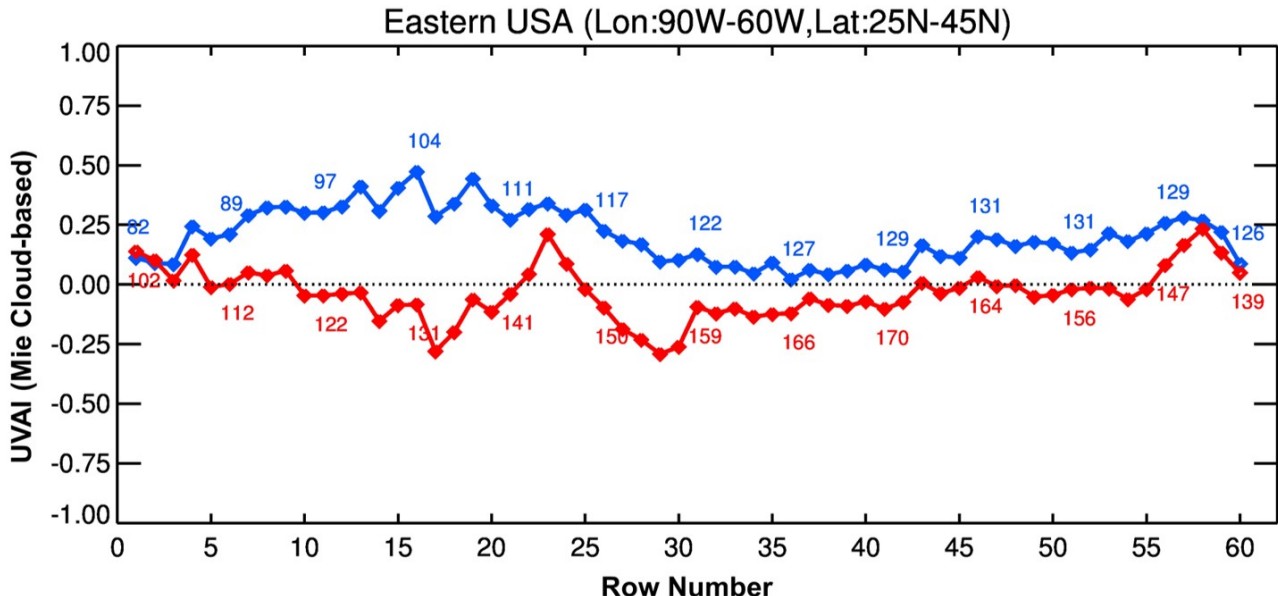

**Figure10. As in Fig. 9for UVAI calculated with a modified algorithm that explicitly accounts for cloud scattering effects.**

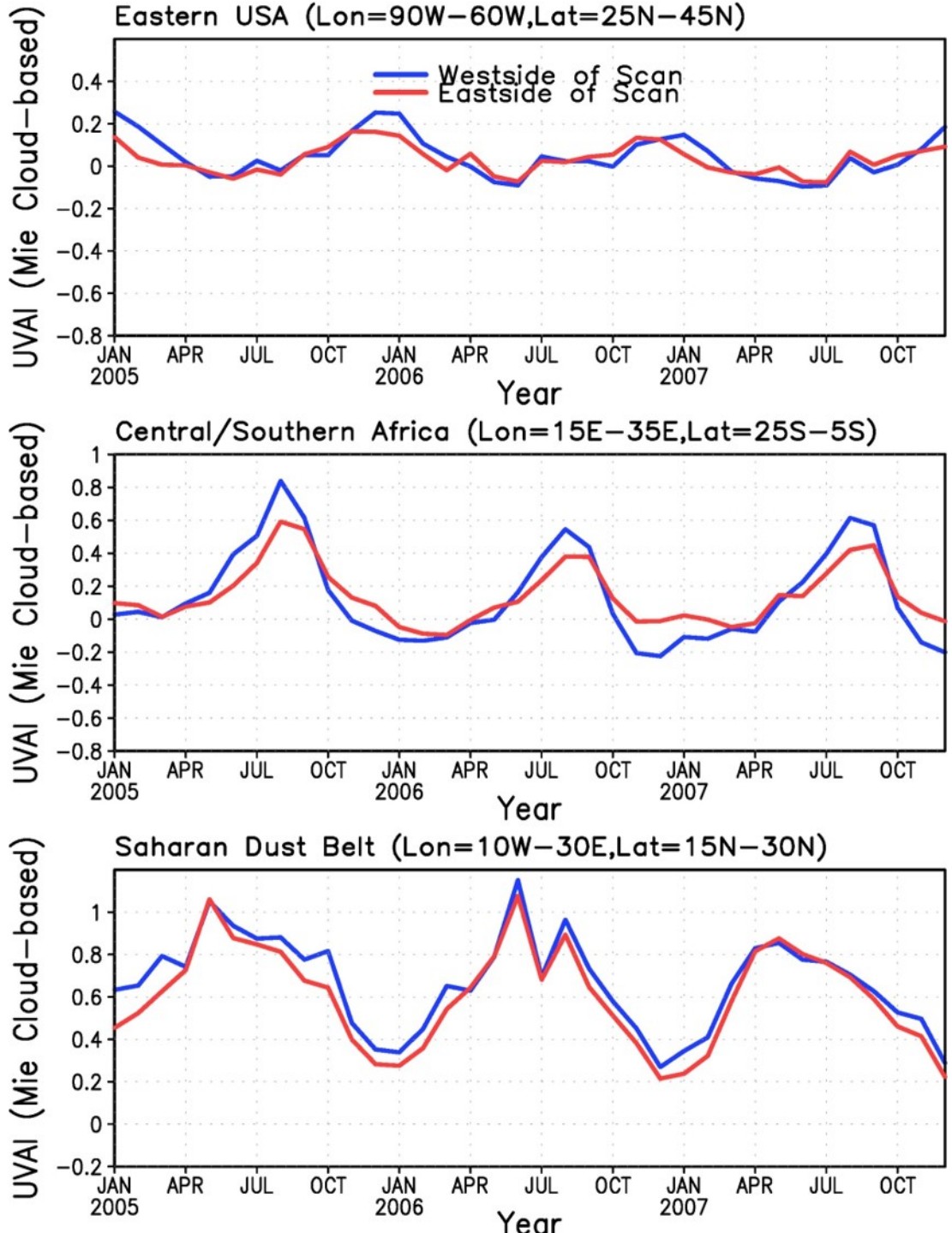

**Figure11. As in Fig. 6 using the Mie UVAI definition.**

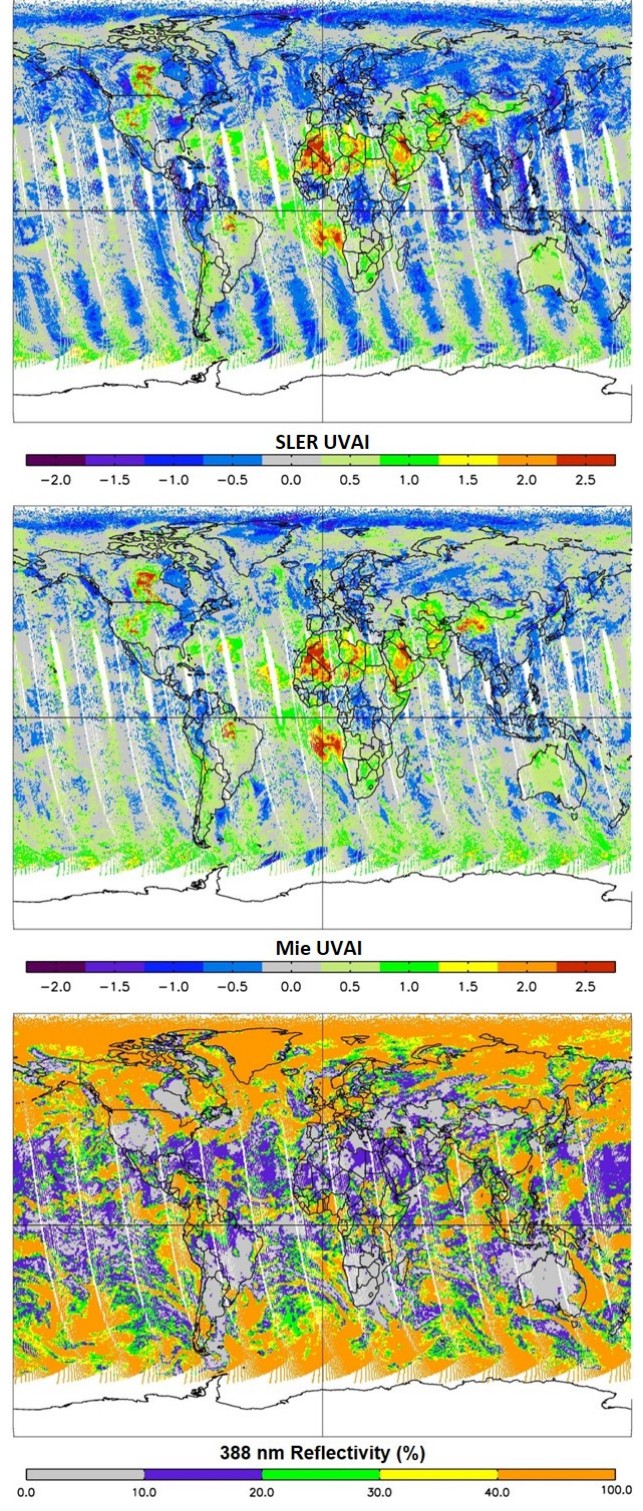

**Fig. 12. Global depiction of SLER AAI, Mie UVAI and scene reflectivity on August 20, 2007.**

## Appendix A

### UV Aerosol Index Calculation

The UVAI [*Herman et al.,* 1997; *Torres et al.*, 1998] is defined as the difference between the logarithms of the ratio of observed

5  and calculated radiances *(I)* at two near UV wavelengths, $\lambda$ and $\lambda_0$

$$UVAI = -100\left\{\log_{10}\left[\frac{I_\lambda^{obs}}{I_{\lambda_0}^{obs}}\right] - \log_{10}\left[\frac{I_\lambda^{cal}}{I_{\lambda_0}^{cal}}\right]\right\} \qquad (A\text{-}1)$$

Since by definition, at $\lambda_0$ the terms $I^{cal}$ and $I^{obs}$ are mathematically identical, Equation (A-1) reduces to

$$UVAI = 100\log_{10}\left[\frac{I_\lambda^{obs}}{I_\lambda^{cal}}\right] \qquad (A\text{-}2)$$

In the above expressions, $\lambda_0$ (generally longer than $\lambda$), represents the wavelength at which the scene Lambertian reflectivity

$R$ is calculated using the equation

$$R_{\lambda_0} = \frac{I_{\lambda_0}^{obs} - I_{\lambda_0}^0}{T_{\lambda_0} + S_{\lambda_0}(I_{\lambda_0}^{obs} - I_{\lambda_0}^0)} \qquad (A\text{-}3)$$

15  where $I_{\lambda_0}^0$ is the path radiance, $T_{\lambda_0}$ is the two-way transmittance, and $S_{\lambda_0}$ is the spherical albedo for illumination from below

of a purely molecular atmosphere.

A.1 Simple Lambertian Equivalent Reflector (SLER) approximation

,. The calculated radiance $I_\lambda^{cal}$, is found by using the calculated $R$ in the Lambertian approximation of the radiative transfer equation under the assumption that R is wavelength independent,

$$I_\lambda^{cal} = I_\lambda^0 + \frac{RT_\lambda}{1 - R\,S_\lambda} \qquad \text{(A-4)}$$

The UVAI is then calculated as in equation (A-1). In this approximation, neither clouds nor surface effects are explicitly included in the radiative transfer calculations.

A.2  Modified LER Approximation

In this approximation, surface and cloud reflected radiance ($I_\lambda^s$ and $I_\lambda^C$, respectively) are explicitly accounted for but still assuming a molecular-only atmosphere. The $I_\lambda^s$ terms are calculated using wavelength independent values of surface albedo of 0.08. The $I_\lambda^C$ terms, on the other hand, are modeled by representing the cloud as a reflecting, opaque surface at located pressure $P_t$ (cloud top) determined from existing climatologies and reflectivity ($R_c$) 0.80. In this approach, a wavelength independent cloud fraction is calculated as

$$f = \frac{I_{\lambda_0}^{obs} - I_{\lambda_0}^s}{I_{\lambda_0}^C - I_{\lambda_0}^s} \qquad \text{(A-5)}$$

For $f$ values less than or equal 1.0, $I_\lambda^{cal}$ values are obtained from the expression

$$I_\lambda^{cal} = (1.0 - f)I_\lambda^s + fI_\lambda^C \qquad \text{(A-6)}$$

and UVAI is then calculated from equation (A-1).

If $f > 1.0$, overcast sky conditions are assumed, and the SLER method of calculating $I_\lambda^{cal}$ is used.