# Peer review of "Impact of the Ozone Monitoring Instrument Row Anomaly on the Long-term Record of Aerosol Products"

_Atmospheric Measurement Techniques, 2017_

## Referee Comment (RC1) · Anonymous Referee #1 · 4 Jan 2018

The paper describes the characterization and correction of view-angle dependent OMI retrieval results of AOD, SSA, and UVAI. Particularly in view of the loss of data from certain OMI detector rows due to the so-called row anomaly, a dependence of retrieval results on view angle (or row number) causes biases in temporal and spatial averages. Torres and co-workers identified the spherical particle assumption to be the reason for the observed view-angle dependency of AOD and SSA retrieved over desert and were able to strongly decrease the bias by using phase functions more appropriate for mineral dust. The UVAI view-angle dependency was found to be mainly caused by the commonly used approximation of clouds as opaque LER surfaces. The UVAI bias over regions affected by clouds could be strongly reduced by adapting the UVAI algorithm

to incorporate a more realistic cloud parameterization.

This manuscript is in a very good condition: the results are impressive and well presented, and the conclusions are of importance to the scientific community, particularly to users of OMI data. My recommendation to the editor is to publish the manuscript as soon as the minor and technical comments below have been addressed in a satisfactory way.

**Minor Comments**

1. Several sentences are extremely long and hard to read (e.g., lines 7-10 on page 4). Please read through the manuscript critically and try to make the sentences shorter, thereby improving readability.

2. Please add one or two literature references to phase functions of small spherical particles (Mie Theory)

3. On page 4, line 9, you mention that "the angular variability of the scattering phase function of aerosols and clouds" is the "ultimate driver of the angular distribution of scattered radiation", but that is disregarding Rayleigh scattering, which is also very anisotropic. This is of course taken into account in your RT calculations and should not cause any trouble within your retrieval, which is probably why it is not mentioned here. But the statement as it is given here is inaccurate.

4. On page 5, starting from line 30, the calculation of new dust phase functions is described, but it is kept rather short. Please be more specific, e.g. by mentioning the assumed fraction of non-spherical particles. How realistic is the selected set of parameters? Regarding the results (particularly in Fig. 4), how representative are they, and what happens if you try different fractions of non-spherical particles? Or different shapes? It would be nice to see this analysis for different particle mixtures (like that shown for different SSA), and in the best case a plot with the range of retrieval errors found for all particle mixtures used by the retrieval algorithm.

5. On page 6, lines 5-6, it says "Retrieval errors transition from overestimations to underestimations at about $155°$ scattering angle". But this is not the case for SSA = 0.97. Any thoughts on why this is so?

6. In the last paragraph of Section 5.3 (page 9), the improvement of the modified-Mie UVAI algorithm with respect to the old version is pointed out. However, the positive UVAI artefacts that appear in the Southern part of each orbit appear to have increased in the new version. Can you comment on that?

7. The modified-Mie UVAI algorithm varies from the SLER algorithm in more than one aspect. In keywords: cloud phase function, cloud opacity, cloud height, surface albedo. Although introducing a more appropriate cloud phase function intuitively seems to be responsible for the decrease in view-angle dependence, the other changes may also have an effect. Did you investigate that? Did you compare results from the MLER (as described in the appendix) to the Mie algorithm?

8. In Fig. 8, there is a large difference between the blue line at row number 20 and the red line at row number 0, although the scattering angle is nearly the same. Is this within the statistical error, or could there be another reason?

9. Please improve the readability of the appendix and add some references (e.g. to Herman et al., JGR 1997 / Torres et al., JGR 1998). The term on the right in eq. A-1 is only equivalent to the term in the middle if the calculated and measured radiances at lambda0 are equal. This requirement is mentioned later in the section, so I suggest to split the equations. It might be more useful to replace the description of the MLER algorithm by one of the Mie algorithm, as the MLER is not used in the presented study.

**Technical Corrections**

p.2, l.4 and 17 *global daily* — daily global

p.2, l.16 *row-anomaly* — row anomaly

p.2, l.18 *two-days* — two days

p.2, l.33 *making use of* — consisting of

p.3, l.23 *slow* — slowed

p.3, l.32-33 *no detection* — missing

p.4, l.3 *The NEUS region (...) representative* — The NEUS region (...) is representative

p.4, l.15 *using separately observations* — treating observations East and West of the nadir separately

p.4, l.18 *monthly average* — average monthly *or* monthly averaged

p.4, l.21 *minima* — minimal *or* minimum

p.4, l.21 *take place* — occur

p.4, l.24 *sulphate aerosols is the most commonly observed aerosol type.* — sulfate and secondary organic aerosols are most common.

p.4, l.26 *produce* — produces *or* provides

p.4, l.30 *region from February through September* — region, particularly from February through September

p.4, l.34 *are in good agreement with each other at the annual minima AOD values* — are in good agreement.

p.5, l.1 *Minima* — Smallest

p.5, l.20 *reproduce* — reproduces

p.5, l.22 Move the citation to the end of the sentence, after the term in brackets.

p.6, l.16 *aerosol models in the* — aerosol models as in the

p.6, l.18 *take place* — occur

p.8, l.14 Which water cloud model? C1?

p.8, l.14 *wavelength-dependent refractive index* — Does the refractive index vary so much between lambda and lambda0 that you need to take the wavelength dependence into account?

p.8, l.15 *prescribed top and bottom levels* — What does this mean?

p.8, eq.(1) and following — Put some space between the equation and the equation number. It's confusing.

p.8, l.24-26 The treatment of surface albedo is also an important change.

p.9, l.22 *set* — sets

p.9, l.25 *actual angular scattering* — actual scattering

p.10, l.4 *were* — where

Fig. 6-8 UVAI is written UV-AI in the figures and the caption. In Figs. 6 and 7, the UVAI method is called LER-based, whereas in the text and in the appendix it is abbreviated SLER. Please be consistent.

[Figure]

---

## Short Comment (SC1) · 11 Jan 2018

Dear Authors,

First of all, let me say that I think your manuscript is very interesting to scientists using OMI aerosol data (and possibly others as well, by serving as a warning!). I don't have any important issues with the paper at all, but rather I'd like to address a concern that has been growing in me for the past few years. In that sense, rather than expecting to resolve this issue before publication of your manuscript, I'd like to kick off a more general discussion on the definition of UVAI.

[Figure]

To me, the UVAI is a quantity whose definition is (relatively) simple, and for which only surface pressure and (depending on the used wavelengths) the total ozone column are required as *a priori* information. This, in my opinion, is one of the strengths of the UVAI. This most simple UVAI version is full of artefacts — for example, the viewing angle dependence that you address in your paper. But its advantage is that those data can easily be reproduced by others, modeled using RT calculations, and compared with UVAI from other satellite instruments. This is exceedingly more difficult if input parameters for the UVAI calculation include surface reflection and cloud height databases, and possibly additional information in the future. To me, the UVAI appears to be turning more and more into a retrieved quantity, instead of the Index as it was defined originally.

The obvious solution for this dilemma would be keeping one "original" UVAI version and one "research" edition. As there are several different UVAI versions available anyway (as you know, OMI alone features three different definitions), this would probably not cause too much confusion — as long as everything is well documented. This would benefit the continuity of the UVAI as the longest-standing record of satellite-based aerosol sensing, without standing in the way of progress.

Kind regards,

Marloes Penning de Vries
* * *

---

## Referee Comment (RC2) · Anonymous Referee #2 · 30 Jan 2018

The manuscript of amt-2017-429 by Torres et al. presents an interesting topic in satellite aerosol retrievals: the representation of the angular distribution of scattered light by aerosols and its consequences for satellite retrieval algorithms. The paper is well structured. The ideas are not new, but they are explored straightforward with appropriate data and theoretical (model) considerations. The results are convincing and relevant. The results are important for the further development of aerosol retrieval algorithms, which are under constant development and have to be adapted to increasingly more sophisticated instrumental capabilities. The aerosol products which are treated in this paper are in dire need of improvement, having been developed for instruments that were designed decades ago. Once state-of-the-art products, delivering daily global

aerosol characteristics, they now suffer from increasingly large inaccuraccies as the instruments' spatial resolution and measurement quality increase considerably. This paper presents an excellent example of the problems that are encountered when unadjusted algorithms are applied to a new, more sophisticated instrument like OMI with much more detail in the across-track direction than previous instruments like TOMS, GOME and SCIAMACHY. The problems addressed here will be even more pronounced in the successors of OMI, and the manuscript presents a clear direction for improvement.

The main problem of the paper is the lack of detail and thoroughness. As said above, the paper is well structured in the sense that the ideas are explored clearly, but the text is sometimes careless to the point of being sloppy, and the analysis lacks the detail that is necessary to check the results should this be desired.

The scientific significance warrants prompt publication of the manuscript, after a careful revision of the text. I will give an overview of the problems I encountered, but this is by no means a comprehensive list, and I encourage the authors to critically revise the manuscript and to provide more details about the analyses.

Specific problems:

The analysis was probably prompted by OMI's reduced viewing capabilities, known as the row anomaly. No unequivocal explanation for this problem is known, and the manuscript's title suggests an analysis of at least its consequences. However, a detailed analysis of the angular distribution of aerosol scattering is presented, but not the consequences of the row anomaly. These topics are clearly connected, but the row anomaly is not treated in the manuscript at all, therefore a more suitable title should be provided. A large part of the introduction is dedicated to the row anomaly, but this is not further treated, except for the statement that only data before 2007 is used because of this. In the conclusion section, at least a general discussion of the row anomaly's consequences in view of the angular distribution of aerosols scattering should be given.

In the introduction, section 3, and a few more times in the main text, the measurements of OMI are referred to as 'scanning'. Although this has no consequences for the results and conclusion of the analysis, I suggest that the authors, who are principle investigators in the OMI project, describe the instrument and its capabilities correctly and accurately.

The introduction lacks details. The reader is expected to know everything about the AOD and SSA retrieval in the OMAERUV algorithm. A reference is given, but I think a brief recap of the angular dependence on aerosol scattering and its consequences for the AOD and SSA is in order here. E.g. in the same way as the treatment of the UVAI product, which is more clear and detailed.

Also the difference between phase functions of spherical particles and spheroids are important to understand, in order to interpret the results.

Abstract:
thru -> through
scattering-angle-dependent -> scattering-angle dependent (multiple times, and inconsistently)

main text:
row-anomaly -> row anomaly (multiple times, and inconsistently)
two-days -> two days
worldwide-coverage -> world wide coverage
etc.

p3 slow -> slowed
p4 representative -> is representative
p4 separately -> separate
p6 Retrieval errors transition from overestimations to underestimations at about 155° scattering angle. -> Rephrase

AOD and AOT are both used, please be consistent.

The level of details of the plot is rather low, leaving questions that seem irrelevant but nag because it hampers a thorough check of the results: the monthly mean pictures seem at close inspection to consist of 4 points per month. Is it a running mean? Or a weekly mean? The analyis in Fig. 4 was done for 'A non-spherical polydispersion'. Which one? What fraction? Was it a specific set that improves the data so well as shown in Fig. 5, or is it robust?
* * *

---

## Author Comment (AC1) · 12 Mar 2018

Dear Editor,

Following the review process, we have produced an enhanced version of the paper that addresses the issues raised in the review process. The main additions to the paper are listed below

-The introduction section was expanded to provide a more complete description of the OMAERUV retrieval algorithm. The following text was added:

"The algorithm, based on TOMS (Total Ozone Mapping Spectrometer) heritage, takes advantage of the interaction of molecular scattering and particle absorption in the UV to detect and quantify absorption properties of UV-absorbing particulate such as carbonaceous, desert dust and volcanic ash aerosols [*Torres et al.,* 1998]. The retrieval algorithm relies on forward calculations of angle-dependent upwelling near UV radiances whose accuracy depend on the correct characterization of both molecular and particle scattering. Aerosol scattering depends on particle size, shape and composition. In the OMAERUV algorithm, the aerosol scattering phase function is calculated using Mie Theory, which applies only to spherical particles. Erroneous scattering phase function characterization may produce large errors in retrieved AOD and SSA values [*Gassó and Torres, 2016*]. OMAERUV uses external ancillary data from other A-train sensors for characterization of aerosol type and information on aerosol layer height [*Torres et al.,* 2013]."

-Section 4.2 was expanded to provide a more detailed description of the modelling aspects of scattering by non-spherical particles. This addition includes two additional figures.

"A known difficulty in the treatment of non-spherical particles is the need of prescribing the fraction of non-spherical elements in the polydispersion, as well as making assumptions on the prevailing aspect ratio values. To address those issues, we follow the statistically optimized approach of *Dubovik et al.* [2011] to account for mixtures of spherical and non-spherical particles, as well as mixtures of spheroids of varying Ɛ values as suggested earlier by *Mishchenko et al.,* [1997]. In *Dubovik et al.* [2006, 2011], the aerosol polydispersion is modelled as a mixture of randomly oriented spheroids. Each size bin consists of a size independent distribution of Ɛ ranging from 0.33 to 2.98 which includes flattened oblate spheroids (Ɛ<1), elongated prolate spheroids (Ɛ>1), in addition to spheres (Ɛ=1). The aspect ratio is distributed in 25 bins, with each Ɛ bin having a fixed weight such that the sum of all weights equals unity as shown in Figure 4. This modelling approach, that closely reproduces the laboratory measured single scattering matrices of mineral dust (Feldspar) reported by *Volten et al* [2001], is currently applied in the operational AERONET (AErosol RObotic NETwork) inversion of measured sky radiances [*Dubovik et al.,* 2006]. The resulting spheroid scattering phase function and it sphere-equivalent representation at 388 nm for single scattering albedo 0.9 are shown in Figure 5. Additional calculations (not shown) as a function of aerosol absorption, indicate that in the near UV, the observed sphere-spheroid phase function difference in the 80°-150° scattering angle range is largest for non-absorbing aerosols, and reduces significantly for SSA values 0.82 and lower. "

[Figure]

Figure 4. Spheroids aspect-ratio-weighted distribution          Figure 5. Sphere and Spheroid scattering phase functions

-A discussion section was added to the paper Section 6.0, including two more figures,

Section 6.0 Discussion

[revised manuscript text omitted]

**Reply to Comments by Anonymous Referee #1**

The paper describes the characterization and correction of view-angle dependent OMI retrieval results of AOD, SSA, and UVAI. Particularly in view of the loss of data from certain OMI detector rows due to the so-called row anomaly, a dependence of retrieval results on view angle (or row number) causes biases in temporal and spatial averages. Torres and co-workers identified the spherical particle assumption to be the reason for the observed view-angle dependency of AOD and SSA retrieved over desert and were able to strongly decrease the bias by using phase functions more appropriate for mineral dust. The UVAI view-angle dependency was found to be mainly caused by the commonly used approximation of clouds as opaque LER surfaces. The UVAI bias over regions affected by clouds could be strongly reduced by adapting the UVAI algorithm to incorporate a more realistic cloud parameterization.

This manuscript is in a very good condition: the results are impressive and well presented, and the conclusions are of importance to the scientific community, particularly to users of OMI data. My recommendation to the editor is to publish the manuscript as soon as the minor and technical comments below have been addressed in a satisfactory way.

Minor Comments

1. Several sentences are extremely long and hard to read (e.g., lines 7-10 on page 4). Please read the manuscript critically and try to make the sentences shorter, thereby improving readability.

*As suggested, we have revisited the manuscript to improve readability.*

2. Please add one or two literature references to phase functions of small spherical particles (Mie Theory).

*The section 4.1 of the paper dealing with the issue of small spherical particles has been revisited. References on the subject has been added.*

*Text added to section 4.1:*

The outcome of these comparative analyses suggests that the particle size distributions, spherical shape, and refractive index used in conjunction with Mie Theory for calculating the scattering phase functions of sulphate and carbonaceous aerosols in the OMAERUV algorithm [Torres et al., 2007] adequately reproduce the observed scattering patterns of small spherical particles [*Kaufman et al.,* 1994; *Dubovik et al.,* 1998].

3. On page 4, line 9, you mention that "the angular variability of the scattering phase function of aerosols and clouds" is the "ultimate driver of the angular distribution of scattered radiation", but that is disregarding Rayleigh scattering, which is also very anisotropic. This is of course taken into account in your RT calculations and should not cause any trouble within your retrieval, which is probably why it is not mentioned here. But the statement as it is given here is inaccurate.

*The statement has been rewritten to eliminate the inaccuracy identified by the referee.*

*Text added to section 3:*

The observed angular distribution is associated with the combined effect of the scattering phase functions of Rayleigh scattering, and scattering by aerosol and cloud particles. Unlike Rayleigh scattering, whose scattering phase function can be unambiguously calculated with great accuracy, the scattering phase functions of aerosols and clouds require detailed information on particle size, shape and optical properties of the scattering elements.

4. On page 5, starting from line 30, the calculation of new dust phase functions is described, but it is kept rather short. Please be more specific, e.g. by mentioning the assumed fraction of non-spherical particles. How realistic is the selected set of parameters? Regarding the results (particularly in Fig. 4), how representative are they, and what happens if you try different fractions of non-spherical particles? Or different shapes? It would be nice to see this analysis for different particle mixtures (like that shown for different SSA), and in the best case a plot with the range of retrieval errors found for all particle mixtures used by the retrieval algorithm.

*The discussion on the non-spherical phase function has been expanded to include additional detail on the choice of spherical/non-spherical particles mixtures. In this work we have adopted the analysis of Dubovik et al [2006, 2014] in which a distribution of spherical/non-spherical particles that closely reproduce laboratory measurements of scattering phase function of Feldspar. To our knowledge, the Dubovik et al approach is the most comprehensive analysis that involves actual observations of phase function. Since our purpose was to account for the non-sphericity effect using an accurate approach, we have done so by adopting the well documented approach of Dubovik et al [2006. 2014]. A detailed sensitivity analysis examining variations of the observation-based formulation of Dubovik et al is beyond the scope of this manuscript.*

*See Text added to section 4.2 above*

5. On page 6, lines 5-6, it says "Retrieval errors transition from overestimations to underestimations at about $155°$ scattering angle". But this is not the case for SSA = 0.97. Any thoughts on why this is so?

*The statement cited by the referee makes reference to the resulting AOD retrieval error depicted in the top panel of Fig. 4. The modelled AOD error does indeed transition from overestimation to underestimation at a scattering angle of $155°$ for all SSA values. The reviewer may be referring to the bottom panel of Fig 4, illustrating the SSA retrieval error for particles of varying absorptivity. The angular dependence of the obtained SSA error varies with aerosol absorption as described in the manuscript.*

6. In the last paragraph of Section 5.3 (page 9), the improvement of the modified Mie UVAI algorithm with respect to the old version is pointed out. However, the positive UVAI artefacts that appear in the Southern part of each orbit appear to have increased in the new version. Can you comment on that?

*The Mie-based UVAI definition is about 0.3 larger than the SLER-based definition. This increase moves up the minima UVAI values associated with water clouds from negative (~ -0.3) in the SLER definition to nearly zero in the Mie-based calculation. As a result, the Mie-UVAI UVAI is in general about 0.3 larger everywhere.*

*Text added to section 5.2*

The calculated value is sensitive to the choice of COD for which a value of 10 has been assumed in this work.  Except at high solar zenith conditions, the calculations are insensitive to assumed cloud top and bottom levels. Accounting for the spectral dependence of surface albedo is also an important difference that will affect the magnitude of the calculated radiances and resulting UVAI values. For the COD value used here the resulting Mie-UVAI is generally 0.3 larger than the SLER definition. This difference increases with assumed COD.

7. The modified-Mie UVAI algorithm varies from the SLER algorithm in more than one aspect. In keywords: cloud phase function, cloud opacity, cloud height, surface albedo. Although introducing a more appropriate cloud phase function intuitively seems to be responsible for the decrease in view-angle dependence, the other changes may also have an effect. Did you investigate that?

*After the cloud phase function, the Mie-UVAI is sensitive, in that order, to cloud opacity, surface albedo and cloud height. Increasing the opacity above the adopted value (10) will increase the UVAI, while lowering it will produce a lower UVAI value. The sensitivity to surface albedo varies regionally. It is largest over arid and semi-arid areas where surface reflectance is spectrally dependent and lowest over densely vegetated areas. The sensitivity to cloud height is negligibly small. A brief discussion of these issues has been added to the paper. A more detailed discussion will be given in a forthcoming publication currently in preparation.*

*See answer to comment 6 above*

Did you compare results from the MLER (as described in the appendix) to the Mie algorithm?

*Yes, the comparison was made.*

*Text added to section 5.1*

*The above analysis was also carried out using the Modified LER UVAI definition described in Appendix A (not shown here for the sake of brevity). As with the SLER UVAI definition, a clear, although slightly reduced in magnitude, across track bias effect was observed.*

8. In Fig. 8, there is a large difference between the blue line at row number 20 and the red line at row number 0, although the scattering angle is nearly the same. Is this within the statistical error, or could there be another reason?

*The two lines are representative of summer and winter conditions. Although the scattering angles are similar, there is no reason to expect exact agreement as the geophysical conditions of the observations are different. Factors such as the presence of aerosols above clouds, presence of ice clouds (instead of water clouds), etc could results in different UVAI values.*

9. Please improve the readability of the appendix and add some references (e.g. to Herman et al., JGR 1997 / Torres et al., JGR 1998).

*Done*

The term on the right in eq. A-1 is only equivalent to the term in the middle if the calculated and measured radiances at lambda0 are equal. This requirement is mentioned later in the section, so I suggest to split the equations.

*Done*

It might be more useful to replace the description of the MLER algorithm by one of the Mie algorithm, as the MLER is not used in the presented study.

*Since the Mie Algorithm is central to the focus of the paper, we prefer to keep its description and discussion in the main body of the paper. We also want to keep the MLER definition in the appendix, since it is referred to in the revised version of the manuscript.*

Technical Corrections
p.2, l.4 and 17 global daily— daily global

p.2, l.16 row-anomaly — row anomaly

p.2, l.18 two-days— two days

p.2, l.33 making use of — consisting of

p.3, l.23 slow— slowed

p.3, l.32-33 no detection — missing

p.4, l.3 The NEUS region (...) representative— The NEUS region (...) is representative

p.4, l.15 using separately observations— treating observations East and West of the nadir separately

p.4, l.18 monthly average — average monthly or monthly averaged

p.4, l.21 minima — minimal or minimum

p.4, l.21 take place— occur

p.4, l.24 sulphate aerosols is the most commonly observed aerosol type.
— sulphate and secondary organic aerosols are most common.

p.4, l.26 produce— produces or provides

p.4, l.30 region from February through September— region, particularly from February through September

p.4, l.34 are in good agreement with each other at the annual minima AOD values — are in good agreement.

p.5, l.1 Minima — Smallest

p.5, l.20 reproduce — reproduces

p.5, l.22 Move the citation to the end of the sentence, after the term in brackets.

p.6, l.16 aerosol models in the— aerosol models as in the p.6, l.18 take place
— occur

*All technical corrections listed above have been addressed*

p.8, l.14 Which water cloud model? C1?

*Yes, we refer to the C1 model. References to Deirmendjian [1964, 1969] have been added to document the source of the models and its nomenclature*

p.8, l.14 wavelength-dependent refractive index— Does the refractive index vary so much between lambda and lambda0 that you need to take the wavelength dependence into account?

*Ignoring the spectral dependence of the refractive index introduces departures as large as 0.2 UVAI units at certain scattering angles. Thus, since the data is available we decided to account for it.*

p.8, l.15 prescribed top and bottom levels — What does this mean?

*We refer to the pressure of cloud top and bottom. We rephrased it in the manuscript for clarity.*

p.8, eq.(1) and following — Put some space between the equation and the equation number. It's confusing.

*Done.*

p.8, l.24-26 The treatment of surface albedo is also an important change.

*Yes, it is mentioned in the revised version of the manuscript. See answer to comment 6 above.*

p.9, l.22 set — sets

*Done*

p.9, l.25 actual angular scattering— actual scattering p.10, l.4 were — where

*Done*

Fig. 6-8 UVAI is written UV-AI in the figures and the caption. In Figs. 6 and 7, the UVAI method is called LER-based, whereas in the text and in the appendix it is abbreviated SLER. Please be consistent.

*Revisited as suggested.*

---

## Author Comment (AC2) · 12 Mar 2018

Dear Editor,

Following the review process, we have produced an enhanced version of the paper that addresses the issues raised in the review process. The main additions to the paper are listed below

-The introduction section was expanded to provide a more complete description of the OMAERUV retrieval algorithm. The following text was added:

"The algorithm, based on TOMS (Total Ozone Mapping Spectrometer) heritage, takes advantage of the interaction of molecular scattering and particle absorption in the UV to detect and quantify absorption properties of UV-absorbing particulate such as carbonaceous, desert dust and volcanic ash aerosols [*Torres et al.,* 1998]. The retrieval algorithm relies on forward calculations of angle-dependent upwelling near UV radiances whose accuracy depend on the correct characterization of both molecular and particle scattering. Aerosol scattering depends on particle size, shape and composition. In the OMAERUV algorithm, the aerosol scattering phase function is calculated using Mie Theory, which applies only to spherical particles. Erroneous scattering phase function characterization may produce large errors in retrieved AOD and SSA values [*Gassó and Torres, 2016*]. OMAERUV uses external ancillary data from other A-train sensors for characterization of aerosol type and information on aerosol layer height [*Torres et al.,* 2013]."

-Section 4.2 was expanded to provide a more detailed description of the modelling aspects of scattering by non-spherical particles. This addition includes two additional figures.

"A known difficulty in the treatment of non-spherical particles is the need of prescribing the fraction of non-spherical elements in the polydispersion, as well as making assumptions on the prevailing aspect ratio values. To address those issues, we follow the statistically optimized approach of *Dubovik et al.* [2011] to account for mixtures of spherical and non-spherical particles, as well as mixtures of spheroids of varying Ɛ values as suggested earlier by *Mishchenko et al.,* [1997]. In *Dubovik et al.* [2006, 2011], the aerosol polydispersion is modelled as a mixture of randomly oriented spheroids. Each size bin consists of a size independent distribution of Ɛ ranging from 0.33 to 2.98 which includes flattened oblate spheroids (Ɛ<1), elongated prolate spheroids (Ɛ>1), in addition to spheres (Ɛ=1). The aspect ratio is distributed in 25 bins, with each Ɛ bin having a fixed weight such that the sum of all weights equals unity as shown in Figure 4. This modelling approach, that closely reproduces the laboratory measured single scattering matrices of mineral dust (Feldspar) reported by *Volten et al* [2001], is currently applied in the operational AERONET (AErosol RObotic NETwork) inversion of measured sky radiances [*Dubovik et al.,* 2006]. The resulting spheroid scattering phase function and it sphere-equivalent representation at 388 nm for single scattering albedo 0.9 are shown in Figure 5. Additional calculations (not shown) as a function of aerosol absorption, indicate that in the near UV, the observed sphere-spheroid phase function difference in the 80°-150° scattering angle range is largest for non-absorbing aerosols, and reduces significantly for SSA values 0.82 and lower. "

[Figure]

Figure 4. Spheroids aspect-ratio-weighted distribution      Figure 5. Sphere and Spheroid scattering phase functions

-A discussion section was added to the paper Section 6.0, including two more figures,

Section 6.0 Discussion

[revised manuscript text omitted]

Reply to Comments to review by Anonymous Referee #2

The manuscript of amt-2017-429 by Torres et al. presents an interesting topic in satellite aerosol retrievals: the representation of the angular distribution of scattered light by aerosols and its consequences for satellite retrieval algorithms. The paper is well structured. The ideas are not new, but they are explored straightforward with appropriate data and theoretical (model) considerations. The results are convincing and relevant. The results are important for the further development of aerosol retrieval algorithms, which are under constant development and have to be adapted to increasingly more sophisticated instrumental capabilities. The aerosol products which are treated in this paper are in dire need of improvement, having been developed for instruments that were designed decades ago. Once state-of-the-art products, delivering daily global aerosol characteristics, they now suffer from increasingly large inaccuracies as the instruments' spatial resolution and measurement quality increase considerably. This paper presents an excellent example of the problems that are encountered when un-adjusted algorithms are applied to a new, more sophisticated instrument like OMI with much more detail in the across-track direction than previous instruments like TOMS, GOME and SCIAMACHY. The problems addressed here will be even more pronounced in the successors of OMI, and the manuscript presents a clear direction for improvement.

The main problem of the paper is the lack of detail and thoroughness. As said above, the paper is well structured in the sense that the ideas are explored clearly, but the text is sometimes careless to the point of being sloppy, and the analysis lacks the detail that is necessary to check the results should this be desired. The scientific significance warrants prompt publication of the manuscript, after a careful revision of the text. I will give an overview of the problems I encountered, but this is by no means a comprehensive list, and I encourage the authors to critically revise the manuscript and to provide more details about the analyses.

*Following the referee's suggestions, the text has been critically revised. As a result, the revised manuscript includes a more detailed discussion of the radiative transfer modelling of non-spherical particles, as well as an extended and discussion of results.*

Specific problems:
The analysis was probably prompted by OMI's reduced viewing capabilities, known as the row anomaly. No unequivocal explanation for this problem is known, and the manuscript's title suggests an analysis of at least its consequences. However, a detailed analysis of the angular distribution of aerosol scattering is presented, but not the consequences of the row anomaly. These topics are clearly connected, but the row anomaly is not treated in the manuscript at all, therefore a more suitable title should be provided.

*The title of the paper has not been changed. We have instead added an extended discussion section that includes the row anomaly effects in the context of the identified scattering modelling issues.*

 A large part of the introduction is dedicated to the row anomaly, but this is not
further treated, except for the statement that only data before 2007 is used because of this. In the conclusion section, at least a general discussion of the row anomaly's consequences in view of the angular distribution of aerosols scattering should be given.

*The added discussion section addresses these issues.*

In the introduction, section 3, and a few more times in the main text, the measurements of OMI are referred to as 'scanning'. Although this has no consequences for the results and conclusion of the analysis, I suggest that the authors, who are principle investigators in the OMI project, describe the instrument and its capabilities correctly and accurately.

*The text has been revised accordingly.*

The introduction lacks details. The reader is expected to know everything about the AOD and SSA retrieval in the OMAERUV algorithm. A reference is given, but I think a brief recap of the angular dependence on aerosol scattering and its consequences for the AOD and SSA is in order here. E.g. in the same way as the treatment of the UVAI product, which is more clear and detailed.
*The description of the retrieval algorithm in the introduction has been expanded emphasizing the angular dependence of particle scattering, and the need to characterize accurately particle size, shape, and composition to avoid errors in retrieved AOD and SSA.*

Also the difference between phase functions of spherical particles and spheroids are important to understand, in order to interpret the results.

*A detailed discussion of the Radiative Transfer aspects of non-spherical particles have been added in section 4.2. Two new figures have been included in this section: Figure 4 describing the distribution of aspect ratio used in the calculation of the scattering phase function of spheroids, and, Figure 5 depicting the resulting spheroid phase function, as well as the originally assumed sphere phase function.*

Abstract:
thru -> through
scattering-angle-dependent -> scattering-angle dependent (multiple times, and inconsistently)
main text: row-anomaly -> row anomaly (multiple times, and inconsistently)
two-days -> two days
worldwide-coverage -> world wide coverage
etc.
p3 slow -> slowed
p4 representative -> is representative
p4 separately -> separate
p6 Retrieval errors transition from overestimations to underestimations at about 155 scattering angle. -> Rephrase
AOD and AOT are both used, please be consistent.

*The above technical corrections have been implemented.*

The level of details of the plot is rather low, leaving questions that seem irrelevant but nag because it hampers a thorough check of the results: the monthly mean pictures seem at close inspection to consist of 4 points per month. Is it a running mean? Or a weekly mean?

*We are assuming the above comment refers to the figure showing regional monthly averages of the analyzed parameters as a function of time for the initial OMI three-year period.*

*We are puzzled about the referee's estimate of 4 as the number of points per month. Each monthly mean value is the result of the averaging of 1 orbit per day (at least) X 30 rows per across-track segment X 60 across-track segments per orbit X 30 days per month = 54,000 points per month. This would be the number of points per month used in the UVAI analysis, which requires no cloud screening. For the retrieved parameters (AOD and SSA), the number of points per monthly mean value reduces to about 10,000.*

The analysis in Fig. 4 was done for 'A non-spherical polydispersion'. Which one? What fraction? Was it a specific set that improves the data so well as shown in Fig. 5, or is it robust?

*Details of the non-spherical poly-dispersion are given in section 4.3 of the revised manuscript.*

---

## Author Comment (AC3) · 12 Mar 2018

Reply to Short Interactive Comment by M. Penning de Vries marloes.penningdevries@mpic.de

Dear Authors,

First of all, let me say that I think your manuscript is very interesting to scientists using OMI aerosol data (and possibly others as well, by serving as a warning!). I don't have any important issues with the paper at all, but rather I'd like to address a concern that has been growing in me for the past few years. In that sense, rather than expecting to resolve this issue before publication of your manuscript, I'd like to kick off a more general discussion on the definition of UVAI. To me, the UVAI is a quantity whose definition is (relatively) simple, and for which only surface pressure and (depending on the used wavelengths) the total ozone column are required as a priori information. This, in my opinion, is one of the strengths of the UVAI. This most simple UVAI version is full of artefacts — for example, the viewing angle dependence that you address in your paper. But its advantage is that those data can easily be reproduced by others, modeled using RT calculations, and compared with UVAI from other satellite instruments. This is exceedingly more difficult if input parameters for the UVAI calculation include surface reflection and cloud height databases, and possibly additional information in the future. To me, the UVAI appears to be turning more and more into a retrieved quantity, instead of the Index as it was defined originally. The obvious solution for this dilemma would be keeping one "original" UVAI version and one "research" edition. As there are several different UVAI versions available any-way (as you know, OMI alone features three different definitions), this would probably not cause too much confusion — as long as everything is well documented. This would benefit the continuity of the UVAI as the longest-standing record of satellite-based aerosol sensing, without standing in the way of progress.

Kind regards,

Marloes Penning de Vries

*Marloes,*

*Thanks for your comment.*

*While we agree that the simplicity of computation is an important advantage of the UVAI, we also think that the accuracy of its interpretation as a genuine aerosol signal should not be sacrificed for the sake of simplicity. By accounting for non-aerosol related effects and removing them from the reported values, the science value of the UVAI is greatly enhanced. The UVAI definition introduced in this manuscript accounts for effects of water clouds that yielded negative values in the previous definition. Those negative values were often misinterpreted as signal associated with non-absorbing aerosols. The removal of those cloud-related effects in the new definition, will facilitate the interpretation of the signal associated with non-absorbing particles. Another important upgrade to the UVAI is accounting for that component of the UVAI arising from the wavelength dependent surface effect of certain surface types.*
*We agree with you on the convenience of keeping both the original and improved definitions. In the OMAERUV algorithm, the original SLER UVAI definition is still reported renamed as*

*'residue', whereas the Mie-based UVAI is reported as the UV aerosol index. Detailed documentation of these changes is currently under development.*

---

## Author Response (AR2)

Dear Editor,

All points in the Editor's review have been addressed:

-Degree signs for scattering angles and geographical coordinates have been added as instructed.
-All references are now listed using normal font
-Figure Captions are all corrected for missing space between 'Figure' and  the corresponding number.
-Indicated spelling mistake of 'Krotkov' was corrected.

Omar Torres